



# A New Gridded Offshore Wind Profile Product for US Coasts Using Machine Learning and Satellite Observations

James Frech[1,2], Korak Saha[1,2], Paige D. Lavin[1,3], Huai-Min Zhang[4], James Reagan[2], Brandon Fung[1]

[1]Earth System Science Interdisciplinary Center (ESSIC), Cooperative Institute for Satellite Earth System Studies (CISESS), University of Maryland, College Park, MD, USA, 20740
[2]National Centers for Environmental Information (NCEI), NOAA/NESDIS, Silver Spring, MD, USA, 20910
[3]Center for Satellite Applications and Research (STAR), NOAA/NESDIS, College Park, MD 20740, USA
[4]National Centers for Environmental Information (NCEI), NOAA/NESDIS, Asheville, NC, 28801

*Correspondence to*: James Frech (james.frech@noaa.gov)

**Abstract.** Offshore wind farms are a low-cost, efficient technology for green energy. They deliver significant economic benefits through their manufacturing and operation, and can be readily deployed at scale. Offshore wind also offers a route to opening up access to renewable energy for a global population, 40% of whom live within 100 km of the coast. Presently, offshore wind speed data around wind turbine hub heights are fairly limited, available either through *in situ* observations from wind masts and floating Light Detection and Ranging (lidar) buoys at selected locations or as forecasting-model based output such as from the 2023 National Renewable Energy Laboratory (NREL) National Offshore Wind (NOW-23) and the European Centre for Medium-Range Weather Forecasts (ECMWF) Reanalysis v5 (ERA5). *In situ* wind profiles are very sparse and costly to obtain en masse, whereas satellite-derived 10 m wind speeds have vast coverage at high resolution. In this study, we show the improvement of deploying machine learning techniques, in particular random forest regression (RFR), over conventional methods for accurately estimating offshore wind speed profiles on a high-resolution (0.25°) grid at 6-hourly resolution from 1987 to 2022 using satellite-derived surface wind speeds from the National Oceanic and Atmospheric Administration (NOAA) National Centers for Environmental Information's (NCEI) Blended Seawinds version 2.0 (NBSv2.0) product. We use 276,577 wind profiles from five publicly available lidar datasets over the Northeast US and California coasts to train and validate a RFR model to extrapolate wind speed profiles up to 200 m. A single extrapolation model applicable to the coastal regions of the contiguous US and Hawai'i is developed, instead of site-specific ones attempted in previous studies.

Our RFR outperforms conventional extrapolation methods at the five training stations under cross validation (where each station is held out from the training once and used for validation), especially under conditions of high vertical wind shear and at wind turbine hub heights (~100 m). This model is then tested on two lidar stations that were not used in the training data and profiles from six NOW-23 station locations to evaluate its performance on unseen data. The final model is applied to the NBSv2.0 data from 1987–2022 to create publicly available wind speed profiles over the coastal regions of the contiguous US and Hawai'i on a 0.25° grid, which are shown to outperform NOW-23 and ERA-5 reanalysis at 100 m using a correlated triple collocation method over five years of matchup data (2015–2019). Gridded maps of wind profiles in the marine boundary layer over US coastal waters will enable the development of a suite of wind energy resources and will help stakeholders in their decision making related to wind-based renewable energy development.

**Short Summary**

A machine learning model is developed using lidar stations around the US coasts to extrapolate wind speed profiles up to the hub heights of wind turbines from surface wind speeds. Independent validation shows that our model vastly outperforms traditional





methods for vertical wind extrapolation. We produce a new long-term gridded dataset of wind speed profiles from 20 to 200 m at 0.25°, 6-hourly resolution from 1987 to 2022 by applying this model to the NOAA/NCEI Blended Seawinds product.


## 1 Introduction

By March 2023, the US offshore wind energy potential capacity will have grown to ~53 GW (Musial et al., 2023). This includes already operational projects, wind farms under construction, and those which are in various other stages of development. Planning an offshore wind farm requires finding the optimal location that fulfills different requirements including pricing,

optimized siting, regulation, and grid integration, among others. For these efforts, stakeholders in the wind energy industry need a suite of wind resources, including a wind atlas that will examine the wind at various heights from the ocean surface up to the wind turbine hub heights. A long-term stable database of wind speeds is a particularly pressing need for the wind energy sector, not only at commonly used hub heights of ~100 m (with rotor diameter of ~90 m), but also at higher hub heights of ~140 m to 160 m as continued technological improvements allow for larger wind turbines.

The biggest hindrance of developing such a long-term database is scarcity of accurate measurements of wind speeds at the hub heights, which requires installing meteorological towers around the coastal US. This becomes less cost-effective as newer, larger turbines are developed since the price of measurements increases with height. Buoy-mounted floating Light Detection and Ranging (lidar) instruments are very accurate alternative devices to measure winds at those heights but are equally expensive. However, due to their lower maintenance cost they are commonly used by wind farm developers. As most of these lidar data are

not publicly available due to proprietary reasons, there remains a scarcity of wind speed observations at hub heights. The only data that are publicly available are from a few lidar stations and the 2023 National Renewable Energy Laboratory (NREL) National Offshore Wind (NOW-23) dataset that is based on the Weather Research and Forecasting (WRF) model in and around the coastal US (National Renewable Energy Laboratory, 2020) described in Sect. 2. As the few publicly available lidar stations have limited spatio-temporal coverage and NOW-23 only covers ~20 years, there is a gap in both real-time and long-term wind speed profile

knowledge along the US coasts. Satellite-based products can be utilized to develop wind speed profile gridded datasets with vast coverage and high resolution that can help address this critical database gap. Using the National Oceanic and Atmospheric Administration (NOAA) National Centers for Environmental Information's (NCEI) Blended Seawinds (NBSv2.0) product, we derive vertical wind speed profiles around the US coasts from July 1987 through 2022.

The buoy-based wind speed (hereafter, "surface" wind speed) measurements from the National Data Buoy Center,

maintained by NOAA (National Data Buoy Center, 1971), have been used along with satellite-based surface wind data to simulate winds at the turbine rotor-swept heights, but these studies used either conventional extrapolation techniques or industry accepted wind models like the Wind Atlas Analysis and Application Program (WAsP) and are very region specific (Doubrawa et al., 2015; Optis et al., 2020a; Optis et al., 2020b).

Several studies have estimated wind profiles from the surface up to the turbine rotor-swept heights using various artificial

intelligence/machine learning (AI/ML) techniques but most of these are site-specific case studies, where lidar measurements were used to train and develop the respective models (Mohandes and Rehman, 2018; Bodini and Optis, 2020a; Optis et al., 2021). A study using two years of wind mast and modeled mesoscale data below 80 m from the New European Wind Atlas (NEWA) to extrapolate 102 m wind speeds showed that multiple machine learning methods including linear, ridge, lasso, elastic net, support vector, decision tree, and random forest regression (RFR) outperformed the power law with RFR performing the best with an

increase in coefficient of determination ($R^2$) of 42% over the power law (Basquero et al., 2022). Over a land-based site in China, the RFR outperformed the power law for extrapolating wind speeds at 120, 160, and 200 m (Liu et al., 2023). At four-land based stations in Oklahoma, the RFR outperformed both the logarithmic and power law based extrapolation, improving accuracy by 25%





when trained and validated at the same site and by 17% when using a round-robin approach where cross validation was performed by leaving out one station from training at a time for validation (Bodini and Optis, 2020a). In addition, the RFR was able to
extrapolate wind profiles during low-level jet (LLJ) occurrences at the four land-based stations, showing improved performance over the logarithmic method, which is unable to replicate such events where surface winds decouple from winds aloft (Optis et al., 2021). LLJs are defined by their wind speed gradient inversion within the stable boundary layer and are important resources for offshore wind energy production along with other high vertical wind shear events (Borvarán, Peña, and Gandoin, 2020; Gadde and Stevens, 2021; Doosttalab et al., 2021). In many previous studies, machine learning shows potential to more accurately estimate
wind profiles over conventional methods, allowing for more informed decision-making for wind farm siting.

RFR in particular has shown promise in extrapolating wind profiles, specifically within the offshore environment. At the E05 Hudson North and E06 Hudson South stations equipped with floating lidar buoys, the RFR outperformed the logarithmic formula, a single column model, and the WRF model, with no evidence of decreased model performance under the same round-robin approach between the two buoys 83 km apart (Optis et al., 2021). As such, they suggested that machine learning is promising
for extrapolating 10 m satellite-resolved wind speeds in the relatively homogeneous offshore environment. In addition, they showed that including the difference between air temperature and sea surface temperature as input to the RFR greatly improved the model by quantifying atmospheric stability. Other work at the three German Forschung In Nord- und Ostsee (FINO1, FINO2, and FINO3) mast stations located in the North Sea and the Baltic Sea around Denmark also found the air-sea temperature difference to be an important input to the RFR (Hatfield et al., 2023). While machine learning has been used to improve the wind extrapolation in a
site-specific manner, we are unaware of any past studies that have used it on a large spatial scale covering multiple coasts, as done in our paper. We use RFR in this analysis as it has shown more promise than other machine learning models for this task.

In this study we apply an RFR developed using offshore lidar data to NBSv2.0 satellite-derived blended gap-free sea surface winds to generate a long-term (1987–2022) product of wind speed profiles up to 200 m on a 0.25° grid around the coastal regions of the contiguous US and Hawai'i. Section 2 introduces the data used for training, validation, and testing of the
extrapolation methods, Sect. 3 describes the conventional vertical wind extrapolation methods, Sect. 4 describes the RFR extrapolation model development, Sect. 5 compares the performance of the RFR and conventional methods both overall and specifically for LLJs and high vertical wind shear events, Sect. 6 describes the independent validation of the extrapolation models, Sect. 7 introduces the new wind profile product, NOAAOffshoreWindProfiles-USA (NOSP), and its error estimation at hub heights, and Sect. 8 summarizes our analysis and gives conclusions.


## 2 Data

### 2.1 Lidar Stations

Data from five offshore lidar stations were used to train and validate the models in this analysis (locations shown in Fig. 1). Three stations (E05 Hudson North, E05 Hudson South West, and E06 Hudson South) are freely available from OceanTech
Services/Det Norske Veritas (DNV) under contract to New York State Energy Research and Development Authority (NYSERDA) and are located in the New York Bight Call Areas. The other two stations are on the California coast at Humboldt (Krishnamurthy and Sheridan, 2023a) and Morro Bay (Krishnamurthy and Sheridan, 2023b) and those data are freely available from the Department of Energy-funded Wind Data Hub. All lidar stations provide 10-minute data including surface wind speed, surface wind direction, wind profiles ranging between 40 m and 200 m at intervals of 20 m, surface air temperature, sea surface temperature, and surface
pressure; all of which are considered in this analysis. In total, there are 276,577 10-minute profiles that are used to train and validate the model with 35% of the data coming from Hudson North, 31% from Hudson South, 15% from Hudson South West, 4% from Morro Bay, and 15% from Humboldt. Additional lidar buoy data from the Atlantic Shores Offshore Wind (ASOW) 4 and 6 stations





were used as an independent test set that was not used in the model training or validation. These included 14,531 and 36,659 profiles from ASOW-4 and ASOW-6, respectively, provided at 10-minute intervals. Each of the seven stations provided data over

a different time period within the range of August 2019 to January 2023 (Table 1).

**2.2 NOAA/NCEI Blended Seawinds**

The NOAA/NCEI Blended Seawinds v2.0 product (NBSv2.0) contains 10 m neutral winds and wind stresses globally gridded at a 0.25° spatial resolution dating back to July 1987 at 6-hourly, daily, and monthly resolution. The data from 17 satellites

is blended to create the product, with up to 7 satellites at a given time, enabling the product to delineate extreme wind speeds with higher accuracy than other wind based products (Saha and Zhang, 2022). The data is currently archived at NCEI and is available in both near-real time as well as in a science quality (post-processed) format from the NOAA CoastWatch server. NBSv2.0 is a well-calibrated, uninterrupted, long-term, gap-free, and stable dataset. The 6-hourly data is used here as the input to generate the final gridded wind profile product.


**2.3 Model and Reanalysis Data**

The offshore wind industry widely uses wind profiles from the NREL Wind Integration National Dataset (WIND) Toolkit (Draxl et. al., 2015), which are available around the coastal US at high spatial and temporal resolution. The latest version of this dataset is the NREL NOW-23 reanalysis data (Bodini et al., 2023; 2024). This product implements the WRF numerical weather

prediction model (NWP) to estimate wind profiles up to 500 m for US coastal regions beginning on January 1, 2000. NOW-23 data currently extends through December 31, 2019 for Hawai'i and the North Pacific regions, through December 21, 2022 for the South Pacific region (e.g., offshore of California), and through December 31, 2020 in all other regions. The 2 km horizontal spatial resolution NOW-23 files are available at both 5-minute and 1-hour time resolution through the Open Energy Data Initiative program of the US Department of Energy via their Amazon Web Service (AWS) public data registry page. Another source of long-

term wind speeds, at 10 m and 100 m only, is the European Centre for Medium-Range Weather Forecasts (ECMWF) Reanalysis v5 (ERA5) product, which uses the Integrated Forecast System (IFS) to produce hourly estimates on a 0.25° global grid dating back to 1940. These fields are downloaded using the Climate Data Store (CDS) Application Program Interface.

One year of wind speed profile data (2019) from six offshore NREL locations representing different oceanic regions around the US coasts is also used to initially evaluate the accuracy of the RFR based model. We average the original 5-minute data

into 6-hourly average profiles. Five years of 6-hourly output between 2015 and 2019 are selected from both the NOW-23 and ERA5 reanalysis datasets to further evaluate our wind speed estimates at 100 m, a commonly used hub height for wind turbines. A triple collocation analysis is used to compare both these products to the wind speeds estimated by applying the RFR to NBSv2.0. Additionally, ERA5 2 m air temperature and sea surface temperature are used to generate air-sea temperature differences ($\Delta T$) as input to the RFR when implemented on NBSv2.0, which does not contain any temperature data.





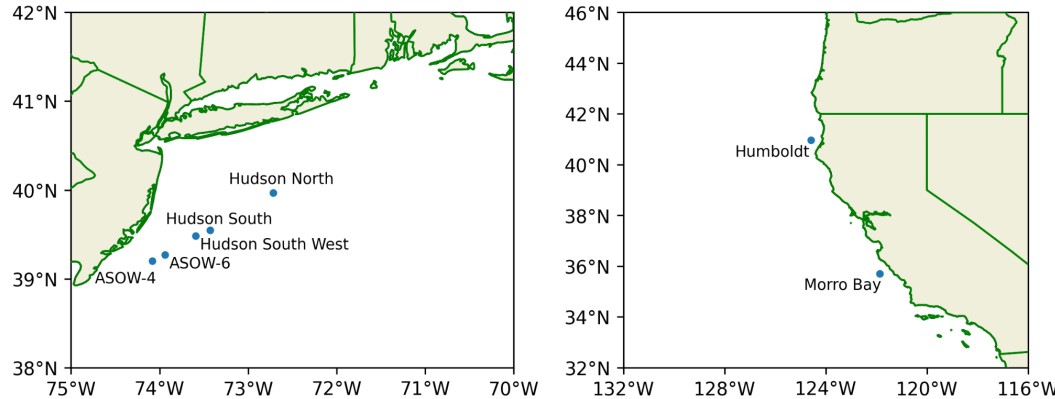

Figure 1: Locations of floating lidar buoy stations used for training, validating, and testing the random forest regression.

| Station | Start/End Dates | Heights of Available Wind Speeds (m) | $N_{obs}$ | Use |
|---|---|---|---|---|
| NYSERDA E05 Hudson North | 8/12/2019 9/19/2021 | 3, 20, 40, 60, 80, 100, 120, 140, 160, 180, 200 | 97779 | train validate |
| NYSERDA E05 Hudson South West | 1/29/2022 1/28/2023 | 3, 20, 40, 60, 80, 100, 120, 140, 160, 180, 200 | 40372 | train validate |
| NYSERDA E06 Hudson South | 9/4/2019 3/27/2022 | 3, 20, 40, 60, 80, 100, 120, 140, 160, 180, 200 | 86860 | train validate |
| Morro Bay, CA | 10/1/2020 2/18/2021 | 4, 40, 60, 80, 90, 100, 120, 140, 160, 180, 200, 220, 240 | 10715 | train validate |
| Humboldt, CA | 10/8/2020 6/29/2022 | 4, 40, 60, 80, 90, 100, 120, 140, 160, 180, 200, 220, 240 | 40851 | train validate |
| Atlantic Shores Offshore Wind 4 | 5/14/2021 9/30/2021 | 10, 40, 60, 80, 90, 100, 120, 140, 160, 180, 200, 250 | 36659 | test |
| Atlantic Shores Offshore Wind 6 | 2/26/2020 5/14/2021 | 10, 40, 60, 80, 90, 100, 120, 140, 160, 180, 200, 250 | 14531 | test |

Table 1: Data availability of each lidar station used in this analysis.

## 3 Conventional Methods for Wind Extrapolation

### 3.1 Logarithmic Law

Conventional physics-based models are typically implemented to vertically extrapolate surface winds, namely a logarithmic law and a power law. The logarithmic law is based on Monin-Obukhov Similarity Theory (Monin and Obukhov, 1954) and relates wind speed $v$ to height $z$ as follows,

$$v(z) = \frac{u_*}{K}\left[ ln\left(\frac{z}{z_0}\right) - \psi\left(\frac{z}{L}\right) + \psi\left(\frac{z_0}{L}\right) \right],$$ (1)





where $u_*$ is the friction velocity of the surface, $K$ is the von Karman Constant (usually 0.4), $z_0$ is the surface roughness, and $\psi$ is a correction function for atmospheric stability that relies on the Obukhov Length ($L$) (Holtslag et al., 2014). $\psi\left(\frac{z_0}{L}\right)$ can usually be ignored as it tends to be minimal compared to $\psi\left(\frac{z}{L}\right)$ in an offshore environment. There are many different formulations for $\psi$, many of which only have applicability within a certain range of $L$. Much research has been done to compare the various different

formulations and create new ones that have their own ranges of applicability (Essa, 1999; Holtslag et al., 2014; Jiménez et al., 2012; Optis et al., 2015; Schlögl et al., 2017). In addition to needing different formulations under certain conditions, the logarithmic law fails to accurately estimate wind profiles in conditions where surface winds decouple from winds aloft, namely in the presence of LLJs (Optis et al., 2021). As such, a more simplistic and accurate model is desired. The neutral logarithmic law removes the stability functions (assumes neutral stability) to give a simpler model but tends to have lower accuracy than other variations of the

logarithmic law. The neutral logarithmic law finds wind speed $v_2$ at height $z_2$ by relating to a reference wind speed $v_1$ at height $z_1$ (Monin and Obukhov, 1954):

$$v_2 = v_1 \frac{ln\left(\frac{z_2}{z_0}\right)}{ln\left(\frac{z_1}{z_0}\right)}, \tag{2}$$

Below, we compare wind profiles extrapolated using the neutral logarithmic law to those from the RFR. Due to lack of variables necessary for estimating the stability functions, namely $u_*$ and $z_0$, we were restricted to using the neutral logarithmic law.


### 3.2 Power Law

The power law for wind profile extrapolation relates wind speeds $v_2$ and $v_1$ at two heights $z_2$ and $z_1$, respectively, as

$$v_2 = v_1 \left(\frac{z_2}{z_1}\right)^\alpha, \tag{3}$$

where $\alpha$ is the wind shear coefficient. When the wind speeds at two heights ($z_1, z_2$) are known, $\alpha$ can be computed directly from

the two wind speeds

$$\alpha = \frac{ln\left(\frac{v_2}{v_1}\right)}{ln\left(\frac{z_2}{z_1}\right)}, \tag{4}$$

which in turn can be used to extrapolate the wind speeds at a third height in the given profile by substituting equation 4 (re-written for height 3 and either height 1 or 2) for $\alpha$ in equation 3.

When wind speed is only provided at one height in a profile, $\alpha$ must be estimated to extrapolate wind speeds at subsequent

heights. $\alpha$ can be estimated as a constant of 0.10 over oceans (Bañuelos-Ruedas 2011). However, $\alpha$ is highly variable over time of day, season, location, wind speed, and height so $\alpha$ should not be used as a constant and instead be modeled as a parameter (Spera and Richards, 1979). Some studies focus on finding the best average estimate of $\alpha$ for a specific wind resource site (Gualtieri and Secci 2011; Werapun et al., 2017). Others define formulations for $\alpha$ that account for the effects of wind speed and surface roughness (Spera and Richards, 1979) or for the effects of atmospheric stability by using correction functions based on Monin-

Obukhov Similarity Theory (Panofsky and Dutton, 1984). While the addition of stability corrections into the formulation of $\alpha$ greatly increases the accuracy of a site-specific long-term average $\alpha$, site-specific information on stability is necessary for this method and it is rather sensitive to the surface roughness $z_0$ (Gualtieri, 2016). A time-varying model for $\alpha$ showed large increases in accuracy over previous models that used a site-specific $\alpha$ (Crippa et al., 2021). However, the model still relies on how $\alpha$ varies around a known specific $\alpha_0$ for a given location or a predetermined constant value. Overall, there is no rule-of-thumb formulation





for $\alpha$ that always best accounts for all of the factors that contribute to variability in $\alpha$. In our power law estimates below (Sect. 5), we use an $\alpha$ value of 0.10 as suggested for the offshore environment (Bañuelos-Ruedas et al., 2011).

In addition, the power law has shown inconsistency when used for estimates of wind energy potential. Power law extrapolation using $\alpha = 1/7$ underestimated wind power potential by approximately 40% (Sisterson et al., 1983). In general, differences in wind energy production estimates when using a power law versus measured energy production may be up to 35%
(Werapun et al., 2017). Global median absolute percentage error in onshore wind turbine capacity factor estimations are as large as 36.9% when using $\alpha = 0.14$ and 5.5% when using mean power law exponents (Jung et al., 2021). As such, more accurate methods of vertically extrapolating wind speeds are critical for accurate representation of wind energy production.

**4 Random Forest Regression (RFR) Model Training**

Random forest regression (RFR) is a machine learning algorithm that takes an ensemble average of the predictions from its members, decision trees, to make one final prediction for each set of input data (Breiman, 2001). Each decision tree is trained on a bootstrapped subset (sampling with replacement) of the full training set and each decision within the tree is made only considering a random subset of the input parameters (i.e., "features") at each split to add variability to the structures of the trees. Both of these model architecture choices add "randomness" to the model. Each split is made by choosing the optimal value of one
of the features available at that "branch" such that the data in that branch is split into two new branches, each with the smallest possible within-group variance. This process is continued until the branch contains a number of observations less than or equal to the hyperparameter (set by the user) for the minimum number of samples required to be at a branch node. Once this minimum sample size is reached the branch is termed a "leaf" and is no longer split. We use the *RandomForestRegressor* function from the *scikit-learn* Python library for this analysis. By training each decision tree in the RFR on diverse subsets of the data and then
averaging their predictions, it both increases accuracy and reduces overfitting.

The RFR model is trained to estimate wind speed profiles from 40 m to 200 m at 20 m intervals. We chose to develop a single model to predict the entire profile to reduce the computation time needed, as compared to training different models for every height, but found identical performance in both cases. Inputs considered in the model for prediction are "surface" (10 m) wind speed (w10), surface wind direction ($\theta$), surface air temperature (T), sea surface temperature (SST), surface pressure, hour of day,
time of year, and the difference between T and SST (T) from the five lidar stations described above. The time of year was calculated as an index for the number of 10 minute intervals (the training data resolution) in a year starting January 1, 00:00:00 and ending December 31, 23:50:00. As the stations do not directly have 10 m wind speed available, w10 was interpolated using the power law with $\alpha$ calculated using the wind shear between the surface buoy wind speed and the next lowest height available (20 m for Hudson stations, 40 m for Morro Bay and Humboldt).The cyclical features (wind direction, hour of day, and time of year) were decomposed
into sine and cosine components to preserve their cyclical nature (i.e., to ensure 11 p.m. is equally close to 10 p.m. as it is to midnight) consistent with the treatment of such variables in previous RFR studies (e.g., Sharp et al., 2022). Only time and surface variables are considered as inputs in our model, but other studies (Liu et al., 2023; Bodini and Optis, 2020a; Baquero et al., 2022) included variables at several heights as inputs in their models. While inputs at other heights could further improve our model, these inputs would be unrealistic for implementation on a gridded wind profile product as no gridded products exist containing observed
wind speeds or other variables at those heights. Additionally, the training data does not contain any profile data other than the wind speeds (and wind direction at the Hudson stations only). The inclusion of other surface variables like friction velocity, the Charnock coefficient, and sensible heat flux have proved to be important features (Liu et al., 2023) and could potentially improve the model further, but these data are not available at the training stations used in this analysis.




A cross validation method in which each station was held out as a validation set for a model trained using the other stations, hereafter leave-one-station-out cross validation (LOSOCV), was implemented during the model training to avoid overfitting to the training stations, consistent with previous studies (Bodini and Optis, 2020a; 2020b). In this LOSOCV approach, five different RFR models were constructed and each time a different one of the five training/validation lidar sites was not included in the training dataset. After training an RFR on the other four sites' data, the input data from the validation site for that model was run through the RFR to assess its performance relative to the observed wind profiles at that location. The goal of this approach is to ensure that each model has no prior knowledge of wind profiles at its respective validation site. While the three Hudson stations may be close enough to one another that some prior knowledge of wind profiles in the area may be known in validating on those stations, the California stations are 631 km apart so it is unlikely that there is prior knowledge of the wind profiles at either station within the model when validating on those stations.

It is important to optimize the input variables selected for the model by removing features that have a negative or negligible effect on the model's accuracy upon inclusion. This will maximize the model's accuracy while minimizing the computation time. To decide which features are important to keep, only features that clearly decrease the errors of the model at all locations are included. Initially, five leave-one-station-out RFR models with all 11 features were trained, each with the intention of validating performance at a hold-out station and assessing overall feature importances. Average values for feature importance across all five models were evaluated during feature selection (Fig. 2). Both w10 and $\Delta T$ had substantially higher feature importances than the other variables so they were immediately selected for inclusion in the final model. While the other features had much smaller feature importances, a forward sequential feature selection process was used to determine if any of the remaining variables further minimized model errors. This process is important as significant cross-correlations between the variables may not be reflected in the feature importances. Additional models were created using w10, $\Delta T$, and one of the remaining features considered one at a time. Both the sine and cosine components of cyclical features were considered as one feature in this process. These model outputs were then compared to the ones that only used w10 and $\Delta T$ as inputs. If a model with three features showed smaller errors compared to the errors of the model with two features, that would indicate the additional feature was worth including in the final model. This process was done recursively to identify all the features that improved the model. However, no other feature further reduced the errors over all stations (not shown here) so the final model only uses w10 and $\Delta T$ as inputs.

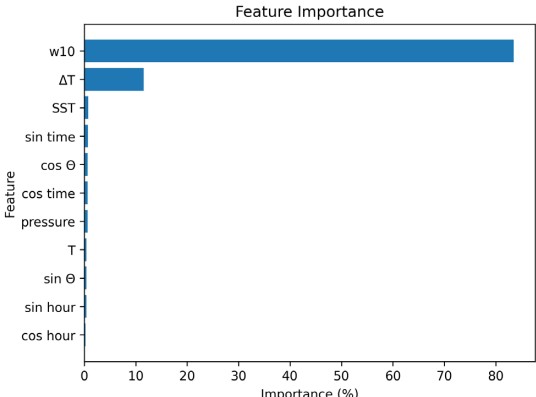

Figure 2: Average feature importance (over the five leave-one-station-out models) of the input variables considered for the random forest regression model.





Once w10 and *ΔT* were chosen as features for the RFR model, the hyperparameters for the model were tuned. The process included tuning three hyperparameters: the number of features considered at each split, the minimum number of observations on each leaf, and the number of trees in the model (nTrees). The number of features to consider at each split could only be one or two as there are only two features in the final model. An essential part of the random forest is that not all features should be considered at each split to prevent the individual underlying decision trees from becoming too similar, which would remove "randomness" from the random forest (Breiman, 2001). Therefore, only one feature was considered for each split in our model. The minimum number of allowed observations on a given leaf is critical to tune because if it is too small it will increase the depth of the trees, increasing the computation time and storage size of the model, while potentially also overfitting the training data. If this hyperparameter is too large, it can result in an overly smoothed model that does not represent all the complexity of the training data. The number of trees in the model was tuned to minimize error in the model and avoid underfitting by averaging over too few trees. The optimal values for these two hyperparameters were determined by analyzing the out-of-bag error (OOB), which corresponds to the average error when all the training observations not included in the bootstrapped subsample used to train a given tree are run through that tree for a pseudo-validation. Minimum leaf size values of 10, 15, 20, 30, 40, and 50 were evaluated for values of nTrees ranging from 20 to 2000 trees incrementing by 20 trees up to 100, followed by increments of 100 trees thereafter. A minimum leaf size of 30 minimized the error and is therefore chosen as the final hyperparameter value. We chose to use nTrees = 1000 trees as this is just above the nTrees value where the OOB error stabilizes and any additional trees would yield no further decrease in error while simply increasing computation time. These values were optimal for all five leave-one-station-out RFR models, which shows that these hyperparameters are not dependent on the locations of the training data.

After selecting the best features and tuning the hyperparameters for our model using our cross-validation process, we trained a final "optimal" RFR model on all five training/validation stations. This allowed us to use as much data as possible in the model in order to improve its accuracy and generalizability. This approach is consistent with other RFR-generated products (e.g., Sharp et al., 2022).

## 5 Extrapolation Model Performance on Training/Validation Data

### 5.1 All wind conditions

Four metrics were employed to assess the skill of our RFR model relative to the conventional physics-based models for wind profile extrapolation in all wind conditions: bias, root mean squared error (RMSE), median absolute error (MAE), and interquartile range of the absolute error (IQR AE). These metrics are obtained by comparing the observed lidar wind speeds and a given model's wind speed predictions (predicted − observed) at every height for each station. Bias is used to determine whether or not the RFR overpredicts or underpredicts on average, RMSE and MAE are used to give estimates of the typical magnitude of the error with MAE being more robust and less sensitive to outliers, and IQR AE determines a spread of the errors around the MAE. For each station and height, these metrics were computed on the validation data from each LOSOCV split for the "optimal" RFR (with w10 and *ΔT* as features and the tuned hyperparameter values) and the "basic" RFR (with all 11 original features and no hyperparameter tuning), as well as for the neutral logarithmic law and the power law using $\alpha = 0.1$ that were applied to all the data (Fig. 3). For the RFR models, the errors are calculated using the LOSOCV approach discussed previously, ensuring the errors are calculated from a model with no prior knowledge of the wind conditions at the given validation site so as to best represent the model's generalization error.

For the majority of the lidar sites, the optimal LOSOCV RFR shows the smallest bias for all heights (Fig. 3a–e). The optimal LOSOCV RFR has negligible bias throughout the profiles at Hudson North and Hudson South with maximum values of 0.07 m s$^{-1}$ and 0.21 m s$^{-1}$, respectively. At the other sites, the bias increases in magnitude with height, increasing from 0.26 m s$^{-1}$



to 0.60 m s⁻¹ at Hudson South West, from −0.10 m s⁻¹ to 0.61 m s⁻¹ at Morro Bay, and from −0.02 m s⁻¹ to −0.59 m s⁻¹ at Humboldt.
The power law at Hudson South West and logarithmic law at Morro Bay are the only two scenarios where the conventional models

show substantially lower biases than the RFR models, ranging from 0.20 m s⁻¹ to −0.01 m s⁻¹ over 40–200 m. While other models
may be slightly less biased at some heights for specific stations (Fig. 3), the optimal RFR has the lowest overall average bias (0.22
m s⁻¹) for all heights and stations followed by the basic RFR (0.25 m s⁻¹) and power law (0.25 m s⁻¹). The neutral logarithmic law
has a larger overall average bias of 0.42 m s⁻¹. In addition, the optimal LOSOCV RFR has substantially lower bias at 200 m than
the basic RFR at Morro Bay (0.25 m s⁻¹ less) and Humboldt (0.18 m s⁻¹ less).

Both the optimal and basic LOSOCV RFRs greatly outperform the power and logarithmic laws in the other metrics,
except at Morro Bay where the difference in performance is less consistent. The optimal RFR has RMSE values increasing with
height from 0.41 m s⁻¹ to 1.44 m s⁻¹ at Hudson North, 0.43 m s⁻¹ to 1.55 m s⁻¹ at Hudson South, 0.48 m s⁻¹ to 1.74 m s⁻¹ at Hudson
South West, 0.34 m s⁻¹ to 1.99 m s⁻¹ at Morro Bay, and 0.57 m s⁻¹ to 2.33 m s⁻¹ at Humboldt (Fig. 3f–j). The average RMSE for
each station's profile is lower for the optimal RFR than for the neutral log law at four sites, decreasing by 48.18% at Hudson North,

48.19% at Hudson South, 44.17% at Hudson South West, and 27.00% at Humboldt. The average percent decrease in RMSE at
Morro Bay has more variability starting from 20.13% at 40 m and decreasing to −4.77% at 200 m, with the neutral log law
producing lower RMSEs at heights of 180 m and above. Compared to the power law, the optimal RFR decreases the profile-
average RMSE by 49.20% at Hudson North, 48.37% at Hudson South, 43.98% at Hudson South West, 20.77% at Humboldt, and
17.09% at Morro Bay. Overall, the optimal RFR has lower RMSEs than the power law at every location and height and at nearly

every location/height compared to the log law.

When compared to the basic RFR, the optimal RFR has slightly higher RMSEs for most heights at the three Hudson
stations, but lower or equivalent RMSEs for all heights at Morro Bay and Humboldt. These differences in RMSE for stations where
the basic RFR has lower RMSEs than the optimal RFR range between 0.05 m s⁻¹ and 0.13 m s⁻¹ at Hudson North, 0.06 m s⁻¹ and
0.17 m s⁻¹ at Hudson South, and 0.01 m s⁻¹ and 0.08 m s⁻¹ at Hudson South West. For stations where the optimal RFR has lower

RMSEs than the basic RFR, the differences range between 0 m s⁻¹ and 0.24 m s⁻¹ at Morro Bay and −0.01 m s⁻¹ and 0.22 m s⁻¹ at
Humboldt. While the basic RFR has slightly lower RMSEs at the Hudson stations, the optimal RFR outperformed the basic RFR
in both bias and RMSE at both Humboldt and Morro Bay. Despite the slightly better RMSE for the basic RFR at the Hudson
stations, it is unclear whether this is only the case due to the stations being close enough to one another that the additional feature
variables may decrease errors in a more localized model. As a higher number of features is less desirable and the optimal model

outperformed the basic model at Humboldt and Morro Bay more than the basic model outperformed the optimal model at the
Hudson stations, we deem that the tradeoff of decreasing the number of features while also lowering errors at Humboldt and Morro
Bay is worth the slight increase in overall RMSE at the Hudson stations.

Similarly to RMSE, both RFR models greatly reduced the MAEs compared to the neutral log and power laws (Fig. 3k–
o). The optimal RFR has MAE values increasing with height from 0.18 m s⁻¹ to 0.56 m s⁻¹ at Hudson North, 0.18 m s⁻¹ to 0.54 m

s⁻¹ at Hudson South, 0.29 m s⁻¹ to 0.84 m s⁻¹ at Hudson South West, 0.21 m s⁻¹ to 0.84 m s⁻¹ at Morro Bay, and 0.27 m s⁻¹ to 1.12
m s⁻¹ at Humboldt. This corresponds to a profile-average percent decrease in MAE from the neutral log law of 63.17% at Hudson
North, 65.25% at Hudson South, 42.18% at Hudson South West, 21.83% at Morro Bay, and 28.57% at Humboldt. Compared to
the power law, the optimal RFR decreases MAE by 72.96% at Hudson North, 74.3% at Hudson South, 55.08% at Hudson South
West, 51.39% at Morro Bay, and 28.2% at Humboldt. Overall, the optimal RFR has lower MAE at every location and height than

the power and neutral log law, except for at 40 m for Morro Bay where the neutral log law marginally beats the optimal RFR.
Similarly to RMSE, for MAE the basic RFR slightly outperforms the optimal RFR at the Hudson stations, whereas the optimal
RFR matches or outperforms the basic RFR at Morro Bay and Humboldt. The basic RFR has lower MAEs than the optimal RFR

with a difference ranging between 0.02 m s⁻¹ and 0.06 m s⁻¹ at Hudson North, 0.02 m s⁻¹ and 0.05 m s⁻¹ at Hudson South, and 0.04 m s⁻¹ and 0.11 m s⁻¹ at Hudson South West. The optimal RFR has lower overall MAEs than the basic RFR with a difference ranging

between −0.02 m s⁻¹ and 0.16 m s⁻¹ at Morro Bay and −0.02 m s⁻¹ and 0.13 m s⁻¹ at Humboldt. As above, we still deem the tradeoff of slightly worse performance at the Hudson stations for less bias and higher accuracy at Morro Bay and Humboldt worthwhile.

The IQR AE values (Fig. 3p–t) are more similar across all profiles at each of the stations with the basic LOSOCV RFRs having the lowest average values at Hudson North (0.63 m s⁻¹), Hudson South (0.66 m s⁻¹), and Hudson South West (0.96 m s⁻¹), the neutral log law having the lowest average values at Morro Bay (0.67 m s⁻¹), and the optimal LOSOCV RFR having the lowest

average IQR AE at Humboldt (1.05 m s⁻¹). Overall, the spread of differences is lower for the RFRs at all stations other than Morro Bay, showing that the RFRs have less variability in their errors than the conventional methods.

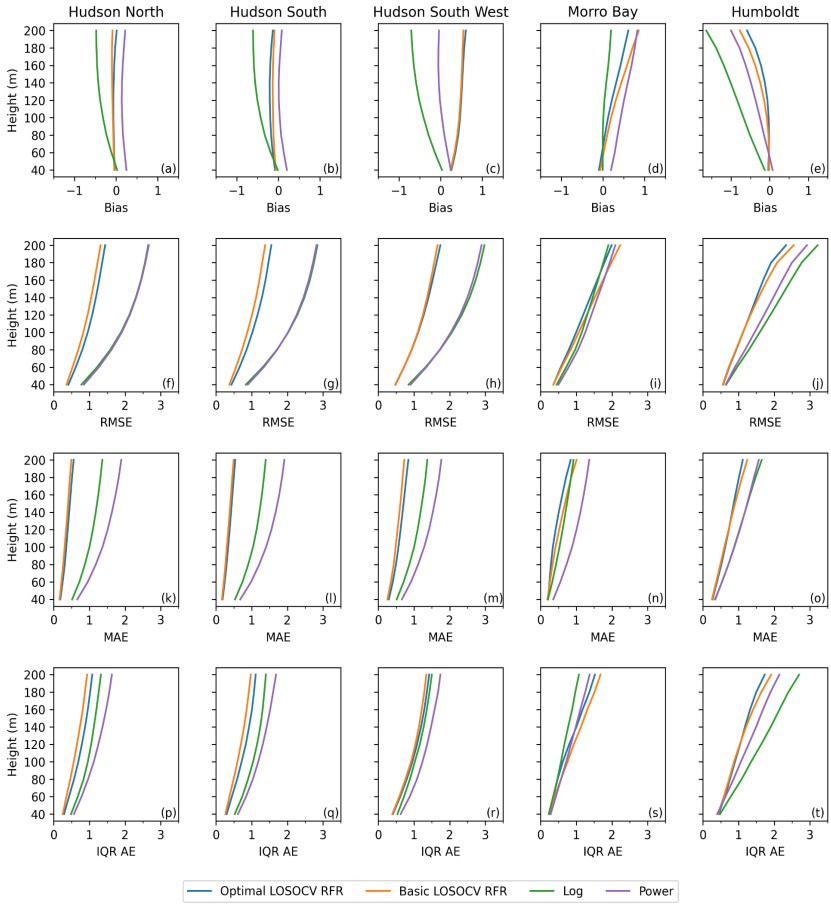

Figure 3: Bias (a–e), root mean squared error (RMSE; f–j), median absolute error (MAE; k–o), and interquartile range of the

absolute error (IQR AE; p–t) profiles at each station for the optimal LOSOCV RFR, basic LOSOCV RFR, neutral log law, and power law models. All plots have units of m s⁻¹.

## 5.2 High Shear Events and Low-Level Jets

As high vertical wind shear events and LLJs both play important roles in wind energy production and load on wind farms

(Borvarán, Peña, and Gandoin, 2020; Gadde and Stevens, 2021; Doosttalab et al., 2021), it is important to accurately model these





phenomena. In this section, we evaluate the performance of the RFR on both LLJs and high shear events and compare with the performance of the conventional models. It is important to note that while LLJs can have a maximum anywhere within the first 1000 m, our training data only reaches to 200 m. Thus, only LLJs with a maximum below 200 m are identified in this analysis.

While the structures of these phenomena are known and can often be distinguished visually (Fig. 4), an exact rigorous
physical criterion to identify these profiles is somewhat elusive. For the purpose of this study, they will be defined in part following the definition used by Debnath et al. (2021) where the 90$^{th}$ percentile in vertical wind speed gradient ($\frac{du}{dz}$ = 0.035 m s$^{-1}$ m$^{-1}$) was used as a threshold for high shear at the Hudson North and Hudson South buoys. While they considered the gradient only between heights within the rotor layer of a turbine (40–160 m), we will consider all heights in our analysis as we are interested in model performance at all heights. For this analysis, LLJs are defined as profiles with a nose (height of wind gradient inversion) below
200 m, $\frac{du}{dz}$ > 0.035 m s$^{-1}$ m$^{-1}$ where $\frac{du}{dz}$ is the vertical wind speed gradient between 10 m and the nose, and a decrease in wind speed from the nose to the top of the profile that is greater than both 1.5 m s$^{-1}$ and 10% of the maximum wind speed (Fig. 4f–j). A high shear event is defined as a profile not already classified as an LLJ and where $\frac{du}{dz}$ > 0.035 m s$^{-1}$ m$^{-1}$ with $\frac{du}{dz}$ calculated as the gradient between 10 m and 200 m (Fig. 4k–o). Any profile that does not fit these criteria are grouped together as "normal" profiles for this analysis (Fig. 4a–e).

Though these LLJ and high shear event definitions may not capture every single profile of these phenomena, they capture the majority and the model errors for each profile type will be representative. Normal profiles account for 82–94% of the data at all training/validation stations, LLJs account for 1–4% of the data at all training/validation stations, and high wind shear profiles account for 8–10% of the data at the Hudson stations, 2% of the data at Morro Bay, and 16% of the data at Humboldt (Fig. 4). The previous error analysis was then repeated separately for each of the three profile types (Fig. 5–7).

All models' performance on the normal profiles is relatively similar to the performance of the models on the full dataset, which is not surprising since the normal profiles comprise > 82% of the data at each site (Fig. 5). For normal profiles, both RFRs still generally outperform the other methods at all stations and heights. Only at Humboldt is the optimal RFR bias less for the normal profiles than for all the profiles combined.



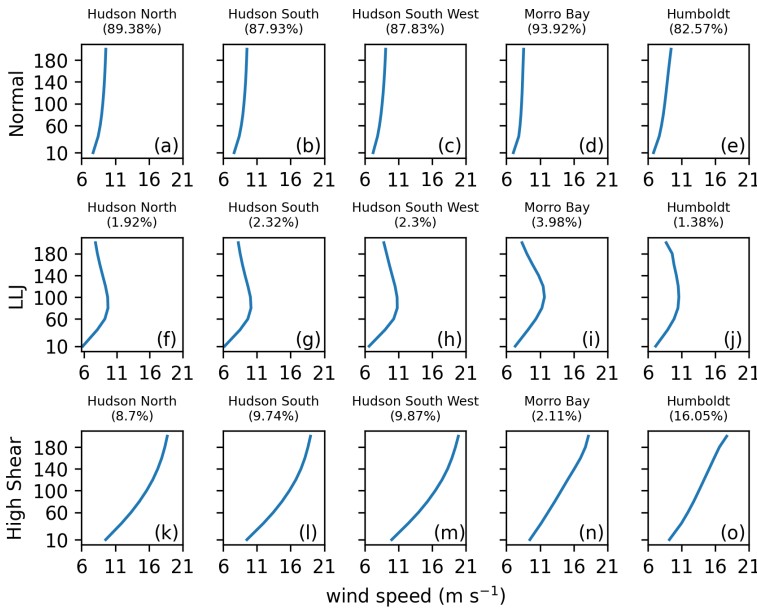

Figure 4: Mean wind speed profiles (m s⁻¹) for normal profiles (a–e), low-level jets (f–j), and high shear events (k–o) at each station. Percentages correspond to the percent of total observations at each station in each group.

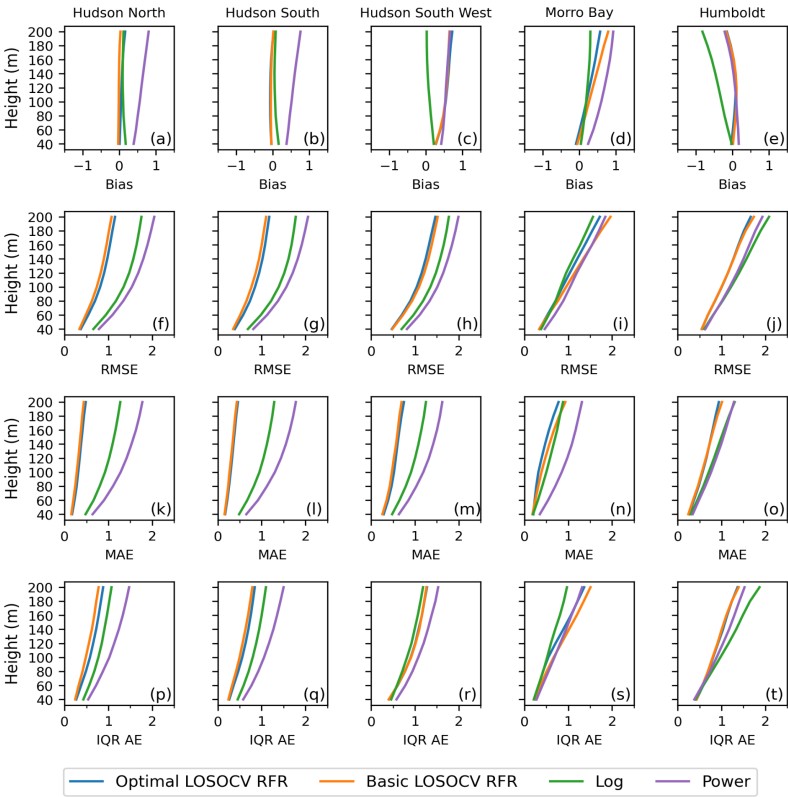

Figure 5: Same as Fig. 3, except metrics only for normal profiles.





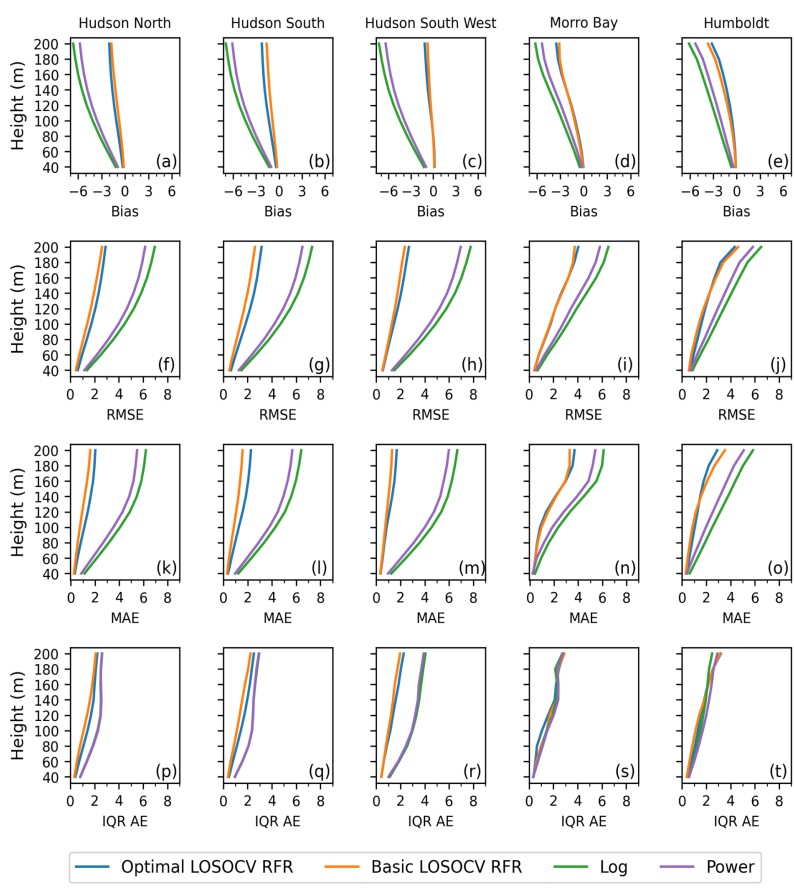


Figure 6: Same as Fig. 3, except metrics only for profiles with high shear events.



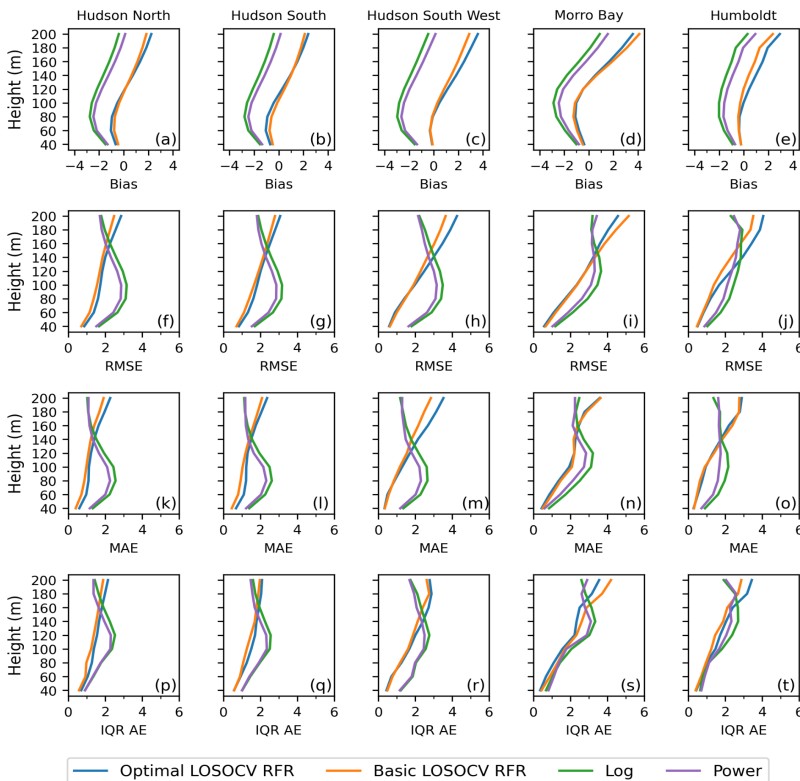

Figure 7: Same as Fig. 3, except metrics only for profiles with low-level jets.


For high shear events, both RFRs vastly outperform the neutral log and power laws (Fig. 6). Compared to the neutral log law, the optimal RFR had an average reduction across all stations and heights of 0.6 m s$^{-1}$ (26.57%) in bias, 2.48 m s$^{-1}$ (53.62%) in RMSE, and 2.64 m s$^{-1}$ (64.4%) in MAE. Compared to the power law, the optimal RFR had an average reduction of 0.64 m s$^{-1}$ (28.93%) in bias, 1.97 m s$^{-1}$ (46.96%) in RMSE, and 2.08 m s$^{-1}$ (55.76%) in MAE. However, while the optimal RFR outperforms

basic RFR when considering all data, the basic RFR seems to either slightly outperform or match the optimal RFR when looking exclusively at high wind shear events. The basic RFR has an average reduction across all stations and heights of 0.14 m s$^{-1}$ (19.3%) in bias, 0.20 m s$^{-1}$ (14.56%) in RMSE, and 0.19 m s$^{-1}$ (21.76%) in MAE. This suggests that the other predictors that did not improve the model trained/validated on the full profile dataset could potentially increase the accuracy of a model trained specifically to predict high wind shear profiles. As the overall errors of the optimal RFR are lower than the basic RFR and significantly less inputs are needed, the optimal RFR is still preferable for our purposes. However, training a separate high wind shear model with additional

features should be considered if those events are of specific interest or are known to dominate regional wind patterns.

For profiles with LLJs, the RFRs generally outperform the other methods below and at wind turbine hub heights, but perform worse at higher heights, especially above 140–160 m (Fig. 7). At 100 m, the optimal RFR has an average reduction across all stations of 2.09 m s$^{-1}$ (82.18%) in bias, 1.24 m s$^{-1}$ (39.20%) in RMSE, and 1.27 m s$^{-1}$ (50.10%) in MAE compared to the neutral

log law. However, at 200 m, the neutral log law has a reduction of 2.46 m s$^{-1}$ (83.48%) in bias, 1.49 m s$^{-1}$ (39.55%) in RMSE, and a 1.49 m s$^{-1}$ (51.52%) reduction in MAE compared to the optimal RFR. Similarly, at 100 m the optimal RFR has an average reduction across all stations of 1.7 m s$^{-1}$ (79.19%) in bias, 0.92 m s$^{-1}$ (31.88%) in RMSE, and 0.88 m s$^{-1}$ (41.08%) in MAE compared to the power law, whereas at 200 m, the power law has an average reduction of 2.37 m s$^{-1}$ (81.92%) in bias, 1.45 m s$^{-1}$ (38.97%)





in RMSE, and 1.44 m s$^{-1}$ (49.36%) in MAE compared to the optimal RFR. It is clear that while the RFR models properly predict

the initial high shear up to turbine hub heights where the conventional methods fail, the current implementation of the RFR models do not have the ability to capture the wind speed gradient inversion above the peak of the LLJs. Conversely, the neutral log and power laws both do not capture the initial high shear of LLJs or the inversion and as a result, only predict the wind speeds above the jet with higher accuracy. As such, the RFR models are preferred for computing wind speeds at hub heights, but may still fall short in energy assessment for LLJs with an inversion layer below 200 m, as determining rotor equivalent wind speeds requires

accurate measurements at all heights within the rotor layer of a turbine. Similarly to high shear events, the basic RFR seems to outperform the optimal RFR slightly on profiles with LLJs, which suggests that training a separate RFR on only the LLJ profiles with additional inputs could increase the accuracy of the RFR for these profiles. This is not done here with our existing NBSv2.0 dataset as it would be impossible to know *a priori* whether the RFR trained on the normal, high shear, or LLJ profiles should be used to estimate the wind profile at a given location when we try to apply the RFR to the NBSv2.0 data to produce our gridded

wind profile product.

        Investigating the distribution of w10 and *ΔT* values for each group of profiles (normal, LLJ, high shear) shows the model's capability to accurately reproduce normal and high wind shear profiles, but only the lower part of the LLJ profiles. After running the w10 and *ΔT* values from the training data back through the final "optimal" RF, we can use the definitions above to classify the predicted profiles and compare those group assignments to those of the observed full profiles across the feature space (Fig. 8).

Many high shear events have a combination of high w10 and strongly positive *ΔT* that is never observed in normal and LLJ profiles. As such, it is encouraging that the RFR model always produces a high shear profile when fed a sufficiently large w10 (> ~7 m s$^{-1}$) and positive *ΔT* (> ~1°C). Observed LLJ and high shear profiles rarely have *ΔT* < −1°C so it is also encouraging that all profiles produced by the RFR using *ΔT* values in this range were normal profiles. However, there is no clear region in the feature space for the model to always predict an LLJ as the LLJ region of the feature space always coincides with that of the normal and/or

high shear groups. This combined with the relatively low amount of LLJ observations compared to normal and high shear profiles may explain why the RFR is not correctly predicting LLJs as the model cannot differentiate them from other profiles with only the given features. For the inversion in an LLJ to be captured, a more complex model and/or other inputs are needed.

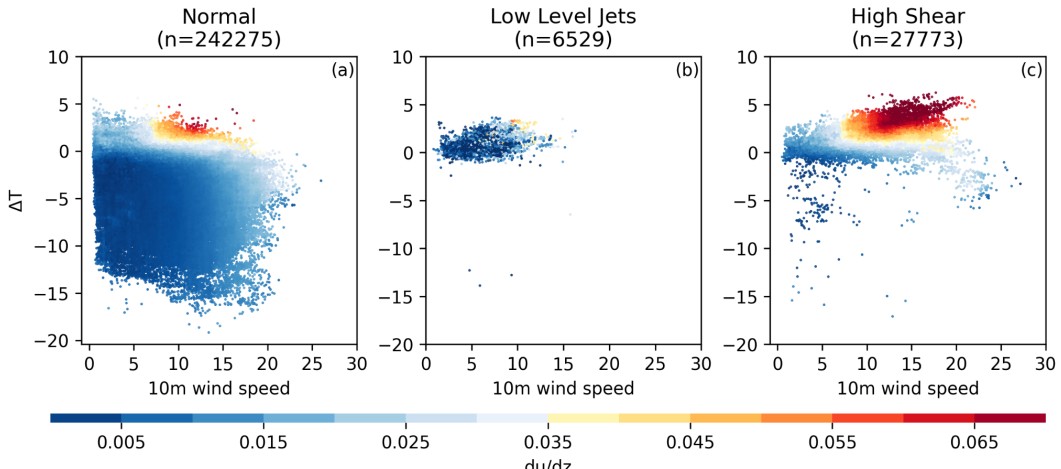



Figure 8: Air-sea temperature difference ($\Delta$T, °C) and 10 m wind speed (w10, m s$^{-1}$) for a) normal, b) low-level jet, and c) high

shear profiles, grouped based on the observed profiles. Wind speed gradients ($\frac{du}{dz}$) based on the RFR estimate of each profile are

shown. The gradient for normal and high shear profiles is calculated between the surface and top of the profiles, while the gradient

for LLJs is calculated between the surface and height of maximum wind speed.

## 6 Extrapolation Model Performance on Test Data

### 6.1 Comparison with Independent Lidar Station Data

The ASOW-4 and ASOW-6 lidar stations were not used during the training and validation of the optimal RFR model and

can therefore be used as a completely independent test dataset to assess the ability of that model to perform on unseen data (i.e., to

"generalize") and to generate initial uncertainties on the RFR-based estimates. The w10 and $\Delta$T values from all profiles at both

stations are extrapolated to full wind profiles from 40–200 m separately by the power law, neutral log law, and optimal RFR models

and then compared to the observed profiles. The performance of the RFR at stations ASOW-4 and ASOW-6 is comparable to the

performance at the five training/validation stations, suggesting the model can perform similarly well at locations it was not trained

on (Fig. 9 and 10).

The ASOW stations are of particular interest as they contain higher percentages of LLJs and high shear events than any

of the training/validation stations (except Humboldt): 5.70% (LLJ) and 14.54% (high shear) at ASOW-4 and 4.43% (LLJ) and

15.58% (high shear) at ASOW-6. 16.05% of profiles at Humboldt were high shear. Together, the LLJ and high shear profiles

account for ~20% of the data at the ASOW stations, showing that accurately predicting these profiles is important for robust

resource assessment at certain locations.

At both ASOW-4 and ASOW-6, the RFR shows considerable improvement over both the neutral log and power laws. For

ASOW-4, the RFR outperforms the other models in all heights for every profile type, except for having larger errors at 180 and

200 m for LLJs where the RFR failed to capture the wind speed gradient inversion (Table 2, Fig. 9), consistent with the above

analysis. At ASOW-6, the RFR still outperforms the conventional methods overall, but is shown to have a bias comparable to the

power law and an MAE comparable to the neutral log law for normal profiles in addition to the higher errors at 180 and 200 m for

LLJ profiles (Table 3, Fig. 10). However, this is not due to a decline in the RFR's performance. Instead, the bias and MAE of the

RFR for normal profiles at ASOW-6 is comparable to those of the conventional methods as the other methods have increased

accuracy at this station compared to at ASOW-4. As such, even when the neutral log and power laws are predicting skillfully at

the given stations, they are still only comparable to the RFR and do not ever substantially outperform the RFR, except for at the

highest heights of the LLJ profiles. In addition, the conventional methods never once have performance anywhere comparable to

that of the RFR on high shear profiles and at turbine hub heights of LLJ profiles. This shows that the RFR still overall has increased

performance over conventional methods at locations independent of training and validation. The error metrics at these two

independent test stations also provide initial estimates of the uncertainty on the RFR-based estimates at other locations independent

of training and validation. For example, at the typical wind turbine hub height (100 m) the RMSE at other independent locations

is likely around 1.4 m s$^{-1}$ (ASOW-4) to 1.8 m s$^{-1}$ (ASOW-6).




|  | $\Delta$Bias | $\Delta$RMSE | $\Delta$MAE | $\Delta$IQR AE |
|---|---|---|---|---|
| RFR−Log (Overall) | −1.06 (−71.16%) | −1.04 (−41.66%) | −0.33 (−29.5%) | −1.42 (−48.10%) |
| RFR−Power (Overall) | −0.79 (−64.85%) | −0.87 (−37.61%) | −0.32 (−28.66) | −1.14 (−42.80%) |
| RFR−Log (Normal) | −0.70 (−79.88%) | −0.61 (−35.29%) | −0.13 (−16.52%) | −0.72 (−37.50%) |
| RFR−Power (Normal) | −0.45 (−71.71%) | −0.50 (−31.15%) | −0.21 (−24.84%) | −0.53 (−30.36%) |
| RFR−Log (LLJ) | −1.25 (−59.4%) | −1.00 (−34.29%) | −0.77 (−35.67%) | −1.10 (−32.82%) |
| RFR−Power (LLJ) | −0.99 (−53.67%) | −0.81 (−29.74%) | −0.60 (−30.2%) | −0.83 (−27.09%) |
| RFR−Log (High Shear) | −2.52 (−54.83%) | −2.26 (−47.77%) | −2.50 (−55.69%) | −2.28 (−44.01%) |
| RFR−Power (High Shear) | −2.17 (−51.13%) | −1.93 (−43.89%) | −2.15 (−51.96%) | −1.97 (−40.36%) |

Table 2: Average change in bias, RMSE, MAE, and IQR AE for the optimal RFR compared to conventional methods for different profile types at ASOW-4. Units are in m s$^{-1}$.

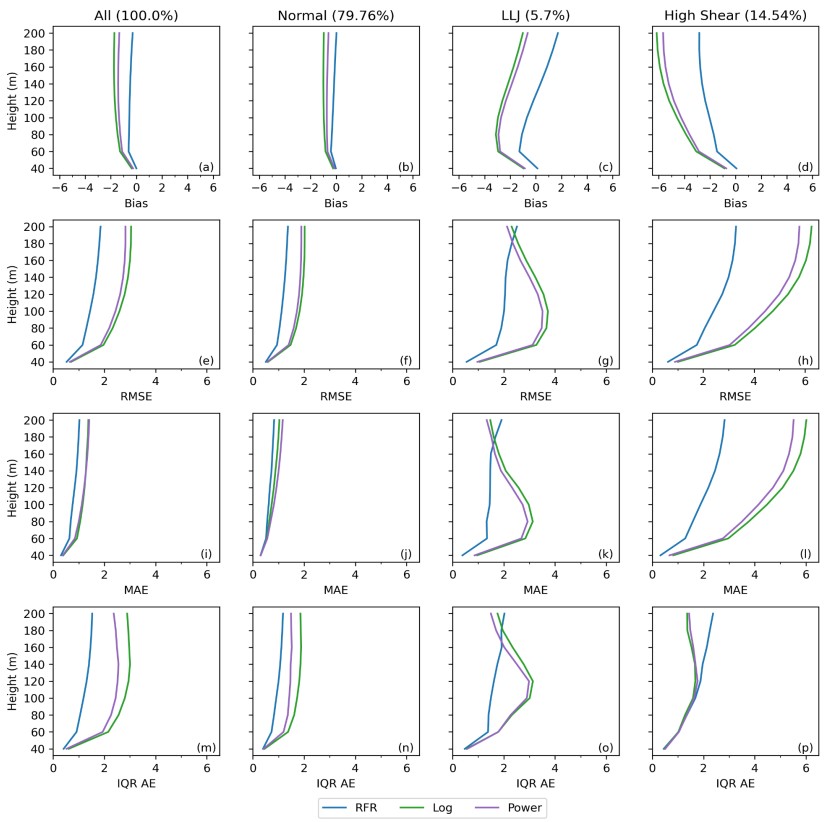



Figure 9: Bias (a–d), root mean squared error (RMSE; e–h), median absolute error (MAE; i–l), and interquartile range of the absolute error (IQR AE; m–p) profiles at the ASOW-4 station for the optimal RFR, neutral log law, and power law models. All plots have units of m s$^{-1}$.

|  | ΔBias | ΔRMSE | ΔMAE | ΔIQR AE |
|---|---|---|---|---|
| RFR−Log (Overall) | −0.76 (−50.42%) | −0.96 (−34.54%) | −0.04 (−4.70%) | −1.05 (−37.98%) |
| RFR−Power (Overall) | −0.43 (−36.48%) | −0.79 (−30.33%) | −0.12 (−11.66%) | −0.78 (−31.34%) |
| RFR−Log (Normal) | −0.26 (−37.80%) | −0.31 (−19.63%) | 0.05 (7.87%) | −0.20 (−12.64%) |
| RFR−Power (Normal) | 0.06 (17.27%) | −0.23 (−15.53%) | −0.07 (−8.26%) | −0.07 (−5.12%) |
| RFR−Log (LLJ) | −1.48 (−61.87%) | −1.16 (−36.56%) | −0.83 (−36.51%) | −1.48 (−38.71%) |
| RFR−Power (LLJ) | −1.22 (−57.29%) | −0.97 (−32.51%) | −0.67 (−31.78%) | −1.23 (−34.45%) |
| RFR−Log (High Shear) | −2.93 (−53.56%) | −2.36 (−40.86%) | −2.93 (−55.90%) | −2.81 (−42.69%) |
| RFR−Power (High Shear) | −2.52 (−49.86%) | −1.98 (−36.74%) | −2.55 (−52.50%) | −2.41 (−39.10%) |

Table 3: Same as Table 2 but for ASOW-6.

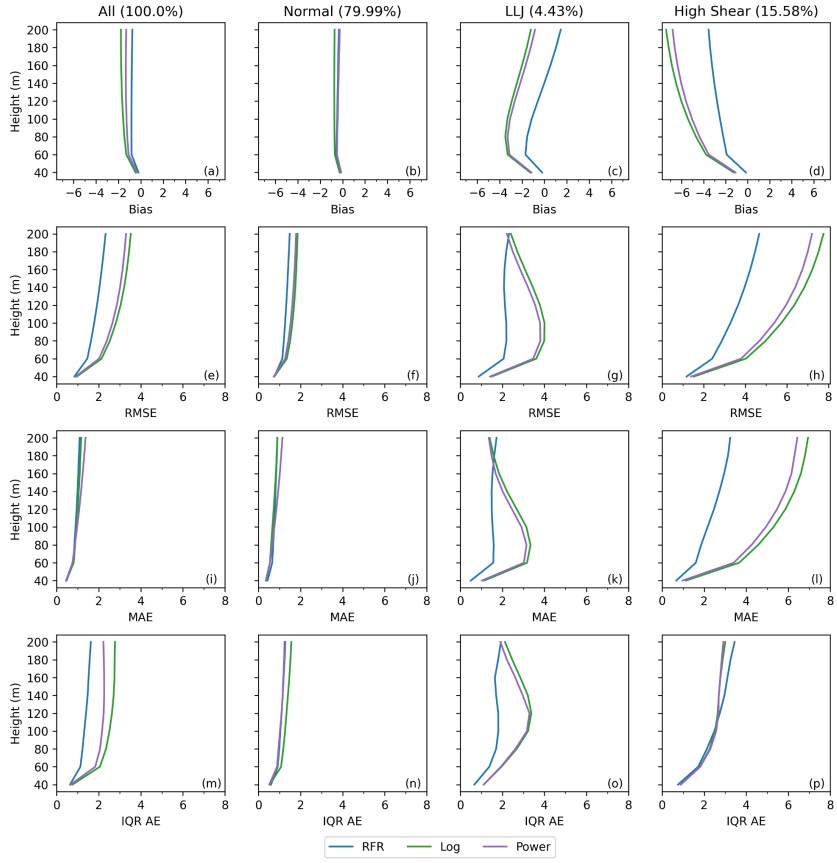

Figure 10: Same as Fig. 9, but metrics now for ASOW-6 station.



### 6.2 Comparison with NREL profiles

NREL NOW-23 wind speed profiles from 2019 at six offshore locations around the US coasts are used to further assess how well the RFR model can perform at stations that it was not trained on (locations provided in Fig. 11). These stations are representative of a number of different important oceanic regions around the coastal US. For this comparison, the RFR-estimated profiles are generated using the w10 and $\Delta T$ from the NOW-23 dataset. This will allow us to directly compare our RFR-estimated wind profiles with NOW-23's WRF-based output to evaluate differences between the extrapolation methods.

To assess the performance of the RFR at these stations, the full profiles from the NOW-23 dataset are used as the "observed" profiles for the error analysis. For most of the stations, the consistent positive bias demonstrates that the RFR consistently overestimates the wind speed at all heights, relative to the NOW-23 profiles (Fig. 12a–f). At station 5 the RFR underestimates the wind speeds from the ground to 140 m and then increasingly overestimates wind speeds from 140 to 200 m. This bias throughout the profile remains within ±1 m s$^{-1}$. The RMSE for all 6 stations remains below 1 m s$^{-1}$ up to turbine hub heights (with exception of station 5), beyond which the RMSEs continue increasing but never go beyond 2 m s$^{-1}$ (Fig. 12 g–l). The MAEs never exceed 1 m s$^{-1}$ and IQR AEs never exceed 1.5 m s$^{-1}$ (Fig. 12 m–x). Overall, when the RFR model uses w10 and $\Delta T$ values from NOW-23 to predict wind speed profiles at six different offshore locations than where the model training data were collected, it does not greatly affect the model's accuracy and the statistics are similar to what was previously seen with lidar stations comparisons (Fig. 3, 9, and 10). This shows that our model can perform skillfully around the coasts of the contiguous US, including regions not included in the training data, such as the Gulf of Mexico, Pacific Northwest, and central East Coast. To our knowledge, no previous studies have shown that an RFR model can successfully extrapolate wind speeds at locations far from the training data.

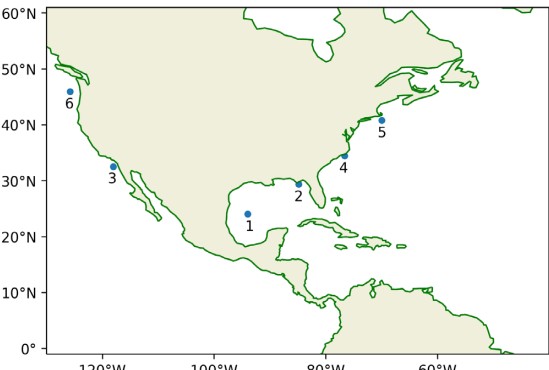

Figure 11: Locations of the six NREL stations used for independent testing of the RFR model.

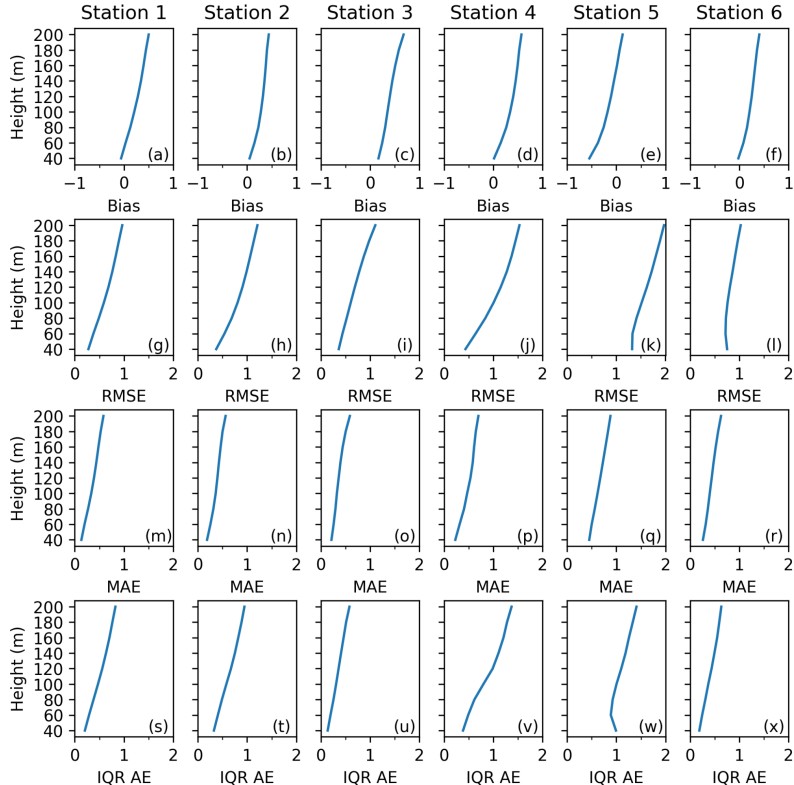

Figure 12: Bias (a–f), root mean squared error (RMSE; g–l), median absolute error (MAE; m–r), and interquartile range of the
absolute error (IQR AE; s–x) profiles at each of the six NOW-23 stations for the profiles extrapolated by the RFR vs. the "observed"
profiles from NOW-23. All plots have units of m s$^{-1}$.

## 7 Application of RFR to NBSv2.0 and Uncertainty Quantification

Once the optimal RFR has been trained, validated, and tested, we apply this model to the NBSv2.0 w10 data and ERA5
$\Delta$T values at 6-hourly resolution over 1987–2022 to generate a long-term wind speed profile product from 20 to 200 m (as well as
the surface value at 10 m) on a 0.25° grid named NOAAOffshoreWindProfiles (NOSP). As the RFR only provides estimation
between 40 to 200 m, these profiles are extended down to 20 m using the power law to interpolate between the 10 m (from
NBSv2.0) and 40 m (from NOSP) wind speeds. The NOSP will be archived for public access. Seasonal climatological wind speeds
are calculated from the NOSP 6-hourly data at three heights (20, 100, and 200 m) to highlight some of the variability that can be
resolved with this product (Fig. 13). Long term mean wind speeds are highest over the subpolar North Atlantic and North Pacific
oceans in all seasons and at all heights, with wind speeds increasing with height over these regions. While the wind speeds over
most of the domain shown decrease in the boreal summer, winds over the California Current System (CCS) are stronger in these
months (consistent with Huyer, 1983), especially at 100 and 200 m. Other than in the CCS region, wind speeds at turbine hub
heights over our domain of interest reach a maximum in boreal winter.



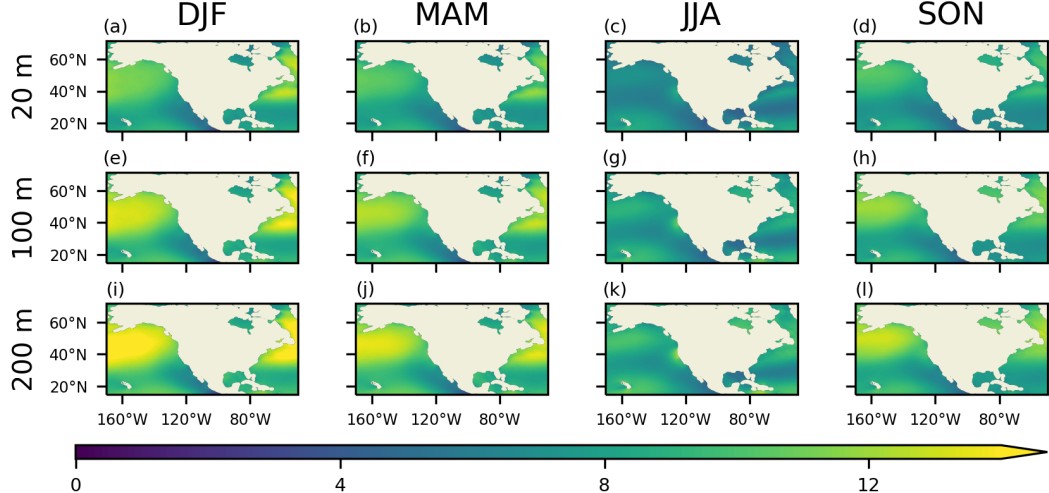


Figure 13: 1987–2022 seasonal climatologies for NOSP extrapolated wind profiles (m s$^{-1}$) at 20 (a–d), 100 (e–h), and 200 m (i–l). DJF = December, January, February. MAM = March, April, May. JJA = June, July, August. SON = September, October, November.

The inherent problem of trying to estimate the uncertainties of the NOSP wind profiles by using the NOW-23 profiles is that although the NOW-23 profiles can be considered as "reference" or "true" values for the comparison, they still have errors as well. Therefore, any statistics for the NOSP profiles will need to be relative to how much error is in the reference profiles. The error in the NOW-23 profiles can originate from many sources including the NWP model (WRF) uncertainties/errors related to the boundary conditions, parametric uncertainty of the model, and errors in input parameters that go into the WRF model, etc. Bodini

et al. (2024) quantifies this uncertainty in the NWP model in terms of bias, centered RMSE, standard deviation and correlation coefficient with respect to both independent lidar as well as NDBC buoy data.

To estimate the error in our product despite these issues, the triple collocation (TC) method is employed. In the TC error analysis, three or more mutually independent datasets can be used to estimate the RMSEs (relative to the unknown "ground truth") of each dataset with good accuracy (McColl et al., 2014; Saha et al., 2020). The basic assumption in this three-way analysis is that

it considers a linear error mode given by Eq. (5), where $X_i$ (with $i$ = 1,2,3) are collocated measurement systems linearly related to the true value of $t$ with $\varepsilon_i$ as additive random errors, $\alpha_i$ and $\beta_i$ as ordinary least-square intercept and slope, respectively, and we estimate the RMSE of $\varepsilon_i$ denoted by $\sigma_{\varepsilon_i}$.

$$X_i = \alpha_i t + \beta_i + \varepsilon_i \,, \tag{5}$$

Another assumption is that all three datasets are mutually uncorrelated ($< \varepsilon_i \varepsilon_j > = 0$) and that they are also uncorrelated

with the "true" value, $t$ ($< t\varepsilon_i > = 0$). McColl et al. (2014) provides an Extended Triple Collocation (ETC) method to estimate the RMSEs for each data along with their sensitivities to the "true" wind speeds. In the case of three datasets with independent errors the RMSEs can be derived using Eq. (6) (Saha et al., 2020) where $\sigma_\varepsilon$ is the RMSE and each $Q$ represents the variance between the two datasets indicated in the subscript:



$$\sigma_\varepsilon = \begin{bmatrix} \sqrt{Q_{11} - \frac{Q_{12}Q_{13}}{Q_{23}}} \\ \sqrt{Q_{22} - \frac{Q_{12}Q_{23}}{Q_{13}}} \\ \sqrt{Q_{33} - \frac{Q_{13}Q_{23}}{Q_{12}}} \end{bmatrix}, \tag{6}$$

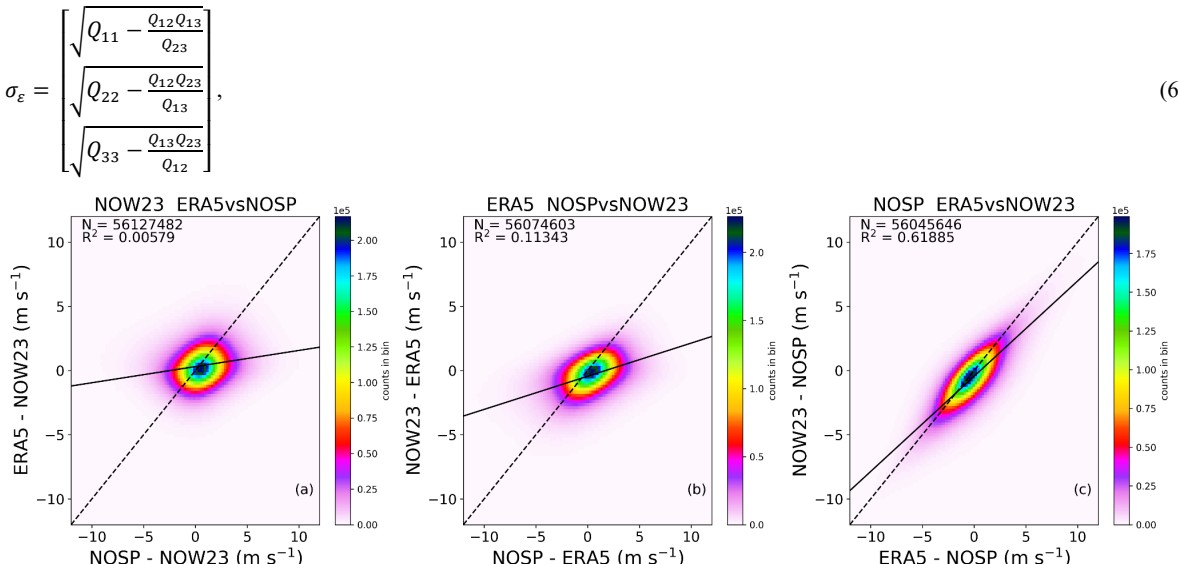

Figure 14: Comparisons in the residual space between a) ERA5 and NOSP, b) NOSP and NOW-23, and c) ERA5 and NOW-23 where the third dataset in each case is used to anchor (i.e., common difference) the two datasets being compared (all in m s$^{-1}$). Each $0.5 \times 0.5$ m s$^{-1}$ bin is colored by the number of matchups in that bin. The solid line represents the linear regression fit and the dashed line is the 1:1 line.

Given the interest in the wind energy sector and the limited availability of independent wind speed datasets, we do this ETC analysis at 100 m only. The three datasets used for this analysis are the NOSP, NOW-23, and ERA5 from 2015 to 2019 at 6-hourly resolution. Initial implementation of Eq. (5) resulted in negative variances, which suggested at least two of the datasets were actually correlated, thereby disregarding one of the key assumptions of the ETC method. To identify which datasets were correlated, three bivariate density (*i.e.*, joint probability) plots in their residual space are generated where two products are compared each time while the third dataset acts as an anchor (common difference) at all of the matchup locations (Fig. 14). With NOW-23 data as the anchor, the $R^2$ between ERA5 and NOSP is ~0.006 while with ERA5 as anchor, the $R^2$ between NOSP and NOW-23 is ~0.113. When NOSP is the anchor, ERA5 and NOW-23 show a very high correlation ($R^2 \approx 0.62$, Pearson correlation coefficient value of ~0.8, and p-value of 0.0). Therefore, it is evident that NOSP is independent from the other two datasets, despite those estimates using $\Delta T$ from ERA5, while ERA5 and NOW-23 are highly correlated. This is likely because the WRF model used to develop the NOW-23 product is initialized and forced at the boundaries with ERA5 data (Rybchuk et al., 2021; Draxl et al., 2021 and Draxl et al., 2015).

Gonzalez-Gambau et al. (2020) provides a new formulation for triple collocation (Correlated Triple Collocation; CTC), for such cases where two out of the three datasets are error-correlated. CTC assumes that the errors between datasets 1 and 2 are correlated, with covariance $< \varepsilon_1\varepsilon_2 > \neq 0$, however they are completely uncorrelated with the error of the third dataset, *i.e.*, $< \varepsilon_1\varepsilon_3 > = 0$ and $< \varepsilon_2\varepsilon_3 > = 0$. For such cases using CTC, RMSEs are given by:

$$\sigma_\varepsilon = \begin{bmatrix} \sqrt{v^2 Q'_{11} + Q'_{22} - Q'_{23}} \\ \sqrt{u^2 Q'_{11} + Q'_{22} - Q'_{23}} \\ \sqrt{Q_{33} - Q'_{23}} \end{bmatrix}, \tag{7}$$



where $u$ and $v$ can be expressed in terms of the variances as $u = \frac{Q_{22} - Q_{12}}{Q_{11} + Q_{22} - 2Q_{12}}$ and $v = \frac{Q_{11} - Q_{12}}{Q_{11} + Q_{22} - 2Q_{12}}$, with $Q'_{11} = Q_{11} + Q_{22} -$

$2Q_{12}$ , $Q'_{22} = u^2 Q_{11} + v^2 Q_{22} + 2uv Q_{12}$, and $Q'_{23} = uQ_{12} + vQ_{23}$. For detailed derivation please refer to appendix A.3 of
Gonzalez-Gambau et al. (2020).

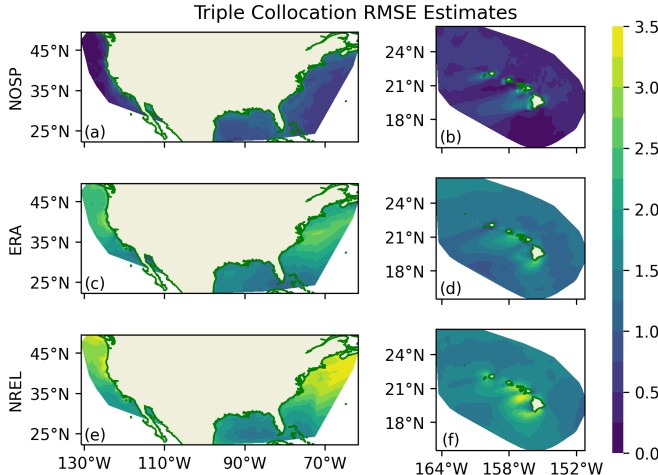

Figure 15: Correlated triple collocation RMSE estimates (m s$^{-1}$) around the coastal regions of the contiguous US and Hawai'i for
the NOSP (a–b), ERA5 (c–d), and NOW-23 datasets (e–f).


Using the CTC formulation, the RMSEs are estimated at each grid point where all three datasets were collocated around
the contiguous US and Hawai'i (Fig. 15). The number of triple collocations (matchups) that are used to estimate these RMSEs are
~56 million, this corresponds to five years of collocated data between the three products. RMSEs are lowest for NOSP (~0.01–2
m s$^{-1}$) followed by ERA5 (~0.17–3.5 m s$^{-1}$) then NOW-23 (~0.65–4.2 m s$^{-1}$). These RMSEs for NOSP are comparable to the

RMSEs calculated above at the two test lidar stations (1.4–1.8 m s$^{-1}$), indicating that the error metrics at those stations were a
reasonable estimation of the RFR's ability to generalize to unseen data. All three products have especially high RMSEs southwest
of Hawai'i coinciding with the unique wind wake found in this region (Xie et al., 2001)

We divided the analysis region further into seven coastal regions of interest for the offshore wind energy sector and
calculated regional-scale RMSEs using the matchups in each region. These seven regions are the North Atlantic Coast (NAC), Mid

Atlantic Coast (MAC), South Atlantic Coast (SAC), Pacific Northwest (PNW), Gulf of Mexico (GoM), Offshore California (OC),
and Hawaiian Coast (HC) (see Fig. 1 in Bodini et al., 2024). The number of matchups between the three datasets at 100 m varies
substantially around the coasts of the contiguous US and Hawai'i (between ~4 to ~13 million; Table 4) due to the varying size of
each region. In all seven regions, NOSP has lower RMSEs (~0.2–1.1 m/s) than the other products (~1.5–3.5 m/s for both the ERA-
5 and NOW-23 data) (Table 4).

Bodini et al., (2024) uses a comprehensive approach of comparing the 20 years of NOW-23 wind speed data at 140 m
with winds extrapolated using a machine learning based model output and reports an uncertainty of below 3 m s$^{-1}$ across the
considered regions. However, the current analysis shows that in regions like the NAC, MAC, and PNW at 100 m the uncertainties
in NOW-23 wind speeds exceed 3 m s$^{-1}$ and it is plausible that the RMSEs could be even higher for higher hub heights.




| Region | RMSEs | | | Total Number of Matchups at 100 m |
|---|---|---|---|---|
| | NOSP | ERA5 | NOW-23 | |
| NAC | 1.11 | 2.14 | 3.35 | 4,148,672 |
| MAC | 0.93 | 2.31 | 3.16 | 4,397,008 |
| SAC | 0.98 | 1.90 | 2.15 | 10,956,000 |
| PNW | 0.29 | 2.41 | 3.10 | 5,375,744 |
| GoM | 1.10 | 1.49 | 1.67 | 13,337,104 |
| OC | 0.71 | 2.24 | 2.37 | 6,303,352 |
| HC | 0.45 | 1.51 | 1.76 | 11,752,136 |

Table 4: RMSEs for all three products (NOSP, ERA5, and NOW-23) and number of triple matchups at 100 m for the seven different regions (North Atlantic Coast (NAC), Mid Atlantic Coast (MAC), South Atlantic Coast (SAC), Pacific Northwest (PNW), Gulf of Mexico (GoM), California Coast (OC), and Hawaiian Coast (HC)) in the coastal US.

## 8 Conclusions

Conventional methods for wind speed profile extrapolation such as the logarithmic and power laws have limitations and greatly underestimate wind power production in many applications. As such, there is a need for new methods of wind speed extrapolation, which led to the use of machine learning for this problem in the past decade. This study focused on building a machine learning model (RFR) to predict wind speed profiles (from 40 m to 200 m above the ocean's surface) around the coastal
regions of the contiguous US and Hawai'i using a gridded satellite-based surface wind speed product (NBSv2.0) as input. This study shows that the RFR algorithm outperforms and is more consistent than the logarithmic and power laws at five lidar stations off the coasts of New York and California when validating using LOSOCV. In addition, the final RFR model requires less input variables (w10 and $\Delta$T) than the other methods to predict vertical wind profiles. The RFR model especially outperforms traditional methods when extrapolating the wind speeds at wind turbine hub heights under conditions of high vertical shear and LLJs. The
only condition where the RFR model did not perform well was above the peak of LLJs as it fails to predict the wind speed gradient inversions that take place there.

    Independent testing of the RFR model using two additional lidar buoys (from the ASOW project) confirms the RFR model's high performance at locations independent of the model's training and its ability to accurately predict profiles with high wind shear. While conventional methods can sometimes approach the accuracy of the RFR for normal profiles, their performance
is much less consistent and never significantly better than the RFR across all of the error metrics. In addition, the ability of the RFR to accurately predict high wind shear makes the model much more useful for wind energy applications than the conventional methods that fail to replicate the high shear. Further independent comparison against profiles from NOW-23 demonstrated the robustness of the RFR as the accuracy of the model does not deteriorate when used to extrapolate wind speeds at locations far from the training sites in New York and California, with errors at the various testing locations (off the Gulf, East, Washington State, and
Southern California coasts) resembling those of the training sites.



Since the training, validation, testing demonstrated that the RFR model can robustly predict wind speeds for the offshore regions of the contiguous US and Hawai'i, it could then be confidently applied to NBSv2.0 6-hourly 0.25°-gridded data to produce 40 m to 200 m wind profiles at this resolution (known as the NOSP). These profile estimates were then extended down to 20 m by applying the power law model between the 10 m (from NBSv2.0) and 40 m (from NOSP) wind speeds.

Lastly, a correlated triple collocation analysis was performed using the NOSP, ERA5, and NOW-23 outputs at 100 m to estimate errors associated with each dataset relative to an unknown ground truth. Across the entire region tested, NOSP consistently had the smallest estimated errors. These results show both the advantages of using satellite-based data over reanalysis and of implementing machine learning versus NWP models for this application.

Since we have demonstrated that the RFR model can robustly predict wind speeds during most conditions found over the
coastal regions of the contiguous US and Hawai'i, future work will continue to improve this model. This includes investigating the use of machine learning for wind extrapolation over larger regions and potentially exploring the use of more complex models. In addition, our RFR model currently lacks the capacity to predict the wind speed gradient inversion of an LLJ, so further research could include identifying other input variables that would be better able to predict these features in an LLJ wind speed profile. Despite these limitations, the RFR model introduced here greatly improves on the conventional methods for extrapolating wind
profiles, particularly over large regions simultaneously. In the future, the NOSP product will be updated to the present and will be produced on a near-real time basis.

**Code Availability:** A package consisting of code involved in developing the model is being pushed to the NOAA/NCEI internal GitLab for code review. Subsequently, the package and the related documentation will be released for the public through the NCEI
archive access.

**Data availability:** The long-term data ranging from 1987–present (the NOAA/NCEI Offshore Seawinds Profiles (NOSP) product), will be archived at NOAA/NCEI and will be served for public use. The NOAA Blended Seawinds surface wind speeds product is available for download at https://oceanwatch.noaa.gov/cwn/products/noaa-ncei-blended-seawinds-nbs-v2.html. NREL NOW-23
data    is    available    at    https://registry.opendata.aws/nrel-pds-wtk/.    ERA-5    reanalysis    is    available    at https://cds.climate.copernicus.eu/cdsapp#!/dataset/10.24381/cds.adbb2d47. Data from lidar stations used in training and validation are    available    from    the    following    sites:    ASOW-4    (https://erddap.maracoos.org/erddap/tabledap/AtlanticShores_ASOW-4_wind.html,    https://erddap.maracoos.org/erddap/tabledap/AtlanticShores_ASOW-4_timeseries.html),    ASOW-6 (https://erddap.maracoos.org/erddap/tabledap/AtlanticShores_ASOW-6_wind.html,
https://erddap.maracoos.org/erddap/tabledap/AtlanticShores_ASOW-6_timeseries.html),    NYSERDA    Hudson    stations (https://oswbuoysny.resourcepanorama.dnv.com/), Humboldt and Morro Bay (https://a2e.energy.gov/project/buoy/data).

**Author Contribution:**

JF and KS conceptualized the idea, performed most of the analysis and coding, and wrote the first draft of the manuscript. PL
provided insight to improve the model and contributed in writing the manuscript. HZ provided valuable input to the final manuscript. JR provided coding support and contributed towards the manuscript. BF helped develop some of the code.

**Competing Interests:**

The authors declare that they have no conflict of interest.




**Disclaimer:**

The scientific results and conclusions, as well as any views or opinions expressed herein, are those of the authors and do not necessarily reflect those of NOAA or the Department of Commerce. Neither NYSERDA nor OceanTech Services/DNV have reviewed the information contained herein and the opinions in this report do not necessarily reflect those of any of these parties.


**Acknowledgements:**

JF, KS, PDL, and BF were supported by the NOAA grant NA19NES4320002 (Cooperative Institute for Satellite Earth System Studies, CISESS) at the University of Maryland/ESSIC. The authors would like to thank the NOAA Center for Artificial Intelligence (NCAI) for supporting the development of this product into both AI- and Cloud-ready formats.

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
