# Peer review of "A New Gridded Offshore Wind Profile Product for US Coasts Using Machine Learning and Satellite Observations"

_Wind Energy Science, 2024_

## Referee Comment (RC1)

**Summary**

This manuscript provides the methodology and analysis behind a newly released publicly available offshore wind dataset called NOST (NOAAOffshoreWindProfiles-USA). The data set is six-hourly at 0.25-degree resolution and spans from 1987 to 2022. The authors implement a machine learning method (random forest regression) to extrapolate wind profiles from satellite-derived surface winds (NOAA's National Centers for Environmental Information's Blended Seawinds version 2.0 (NBSv2.0) product). They train their extrapolation model using lidar data and validate it at the same lidar sites as well as at two other locations whose data was not used for training. They compare their wind profiles with reference wind profiles (log-law, power law) against observations and find that their model outperforms reference wind profiles (as expected). They quantify their errors and also analyze subsets of the data to focus on low-level jets and high-shear conditions. Lastly, they use a triple collocation method to compare the NOSP product with ERA5 and the NOW23 wind dataset and find the lowest estimated errors for NOSP regardless of coastal region. The paper is very well-written, and I believe their dataset would be highly valued by wind farm developers. Overall, this is a great paper that can be improved with a few minor revisions as suggested below.

**Specific Comments**

1.  I would suggest removing or streamlining the first three sentences in the abstract. Considering that this is a wind energy-focused journal, the statements are already widely understood by the journal's target audience.

2.  I do not think that both the log-law and power law need to be used as references. I would recommend just choosing one as they are going to be similar and with similar limitations. Additionally, I could not find the surface roughness value used to generate the log-law profiles. Does the value vary spatially or temporally due to ocean conditions? The authors state that a lack of knowledge regarding $u^*$ and $z\_0$ restrict them to only the neutral log-law, but you still need $z\_0$ for the neutral log-law so I am confused why $u^*$ could not also be determined. I would also think that $u^*$ is also an output of ERA5 similar to the SST.

3.  In general, there are a lot of figures with vertical profiles. I would recommend a few things to improve readability As in 2, I would suggest removing either the log-law or

the power law. RMSE and MAE are also similar metrics. I would recommend just showing one of the two.

4.  In general, the figures need to take advantage of the width of the page to improve readability. They are too small as they currently exist. They also need to be centered. The captions for figures 8 and 9 need to be on the same page as the image.

5.  Did the authors consider changing the shear exponent for the power law for the higher shear cases? I assume that the authors want a model that is independent of parameters that should be tuned; however, using a power law with alpha=0.1 is obviously going to underestimate wind speeds higher up during high-shear conditions.

**Minor Comments**

-   It would improve readability of Tables 2 and 3 if the data subset (overall, normal, etc.) was a separate column
-   Table 2: the entry on the second row for the MAE column is missing a '%'.

---

## Referee Comment (RC3)

This study presents a valuable attempt to address the scarcity of offshore wind speed profile data by developing a machine learning-based extrapolation model utilizing satellite-derived wind data, which could benefit wind resource assessment along U.S. coasts. The authors should be commended for their extensive data collection and model validation efforts across multiple coasts, showcasing the model's performance relative to traditional methods like the Logarithmic and Power Laws. However, while the work offers interesting results, there are notable areas where the study could be improved. Primarily, the claimed novelty of the research is somewhat diluted, as the Random Forest Regression (RFR) model has already been widely applied in similar extrapolation studies, and the spatial and temporal resolution of the resulting dataset is still coarser than other products like NOW23. Additionally, the reliance on standard stability assumptions and shear exponent limitations in the comparison methods may not fully reflect real-world conditions, and the leave-one-out cross-validation (LOSOCV) approach, though thorough, detracts from the goal of developing a single, generalizable model. Furthermore, the study would be strengthened by more focus on the practical application of the new dataset (NOSP) and its insights for wind energy stakeholders, potentially incorporating offshore wind resource assessments and Levelized Cost of Energy (LCOE) analyses. These enhancements, along with restructuring of certain sections and clarifications in the visual elements, could provide a clearer scientific contribution and highlight the value of the NOSP dataset for the wind energy community.

After careful evaluation, I find that the manuscript, in its current form, lacks sufficient novelty to warrant publication. To enhance the paper's contribution, I recommend the authors consider incorporating novel elements that extend beyond current literature, such as using the derived dataset (NOSP) for offshore wind resource assessments or cost analyses (e.g., LCOE calculations) that can directly benefit the wind energy industry. If such advancements are added, the work would more clearly demonstrate its unique value.

Specific comments:

1. With the current pace of growing wind turbine sizes, it is of paramount importance to accurately measure the wind speed profiles, rather than just hub height wind speed. Introduce the importance of wind speed profiles, in terms of their interaction with wind turbines and their impact on loading and power estimation.

2. In addition to tall towers and lidars, sodars also serve as wind profile measuring instruments. Introduce them. Also, lidars have several limitations, such as failed to measure wind speed during raining, snowing, or other precipitation events; no measurement without aerosols. Instead of "lidars are very accurate alternative devices", at lines 54, introduce the limitations as well.

3. Line 59: 20 years of mesoscale model simulated hindcasts of wind profiles (NOW23) can offer a concise resource assessment, which are sufficient in temporal scale, covering annual to decadal analysis, and at a high resolution of 2 km. It doesn't appear to be a gap in the long-term wind speed profile knowledge along the US coast. Rephrase this sentence.

4. Lines 60-64: The authors claim the novelty lies in the idea of bridging critical database gap. However, the NBSv2.0 gridded dataset available at a resolution of 0.25 deg, and at a temporal resolution of 6 hours, which is poorer in temporal and spatial resolution, than NOW23, except for the coverage. Here it dilutes the novelty.

5. There are several studies extrapolated wind speed from surface to 200 m level, utilizing random forest regression (RFR). A very recent study under review is Rouholahnejad et al. (2024), which did exact similar work. The authors could have referred to this article. In addition, the RFR methodology has been adopted in several other studies, showcasing the better predictability in low-level jet cases and high shear events, compared to the conventional low-logarithmic law power law. In comparison to the existing studies, there seems little added value in this study to the scientific community.

Rouholahnejad, F. and Gottschall, J.: Characterization of Local Wind Profiles: A Random Forest Approach for Enhanced Wind Profile Extrapolation, Wind Energ. Sci. Discuss. [preprint], https://doi.org/10.5194/wes-2023-178, in review, 2024.

6. In fact, this current study resembles extension of Optis et al., 2021. However, there is limited scientific addition in novelty.

7. Lines 94-96: the authors state that the main novelty of this present work lie in extrapolating wind speed profiles on a larger spatial scale covering multiple coasts. However, they focused vastly on the RFR model and its validity compared to the conventional methods, which in fact several researchers have already reported. Similar to the previous studies, the current study also failed to accurately estimate the LLJ cases. If the authors have captured the LLJ events, it could be a stand-alone novelty.

8. Lines 143-150: this paragraph seems out of order and introduces confusion, since the ML methodology has not been explained. For a clarity, remove this from here and explain it when introducing the RFR model training.

9. One major drawback of this study is using the conventional Logarithmic Law and Power Law for comparison, which have been proven to be inadequate in extrapolating wind profiles, due to their dependance on stability conditions and shear exponent. Also, the assumption of neutral conditions for Logarithmic Law and a constant sear exponent for Power Law already introduces substantial errors in wind profile

extrapolation. To make the study novel, it is better to consider all the stability conditions and variable shear exponent in estimating wind speed profiles.

10. The authors gave a tremendous importance to the leave-out-station-out cross validation. It is common practice to split the data into training and validation, to optimize the model parameters. Instead, the authors constructed five RFR models by leaving one station at a time, and testing the model accuracies on these stations. This actually comes into "testing" the model accuracy, but not the "validation". The authors should understand that the wind energy community surpassed validating the accuracy of RFR model, rather a unique model accurate enough to extrapolate the entire wind speed profile is needed. Instead of bombarding the manuscript with LOSOCV model, the authors should bring new insights from the one single optimized RFR model.

11. The authors conducted feature importance and came up with only two variables, namely 10 m wind speed and temperature difference. The model trained with these parameters is termed as "Optimal LOSOCV RFR", while the model trained with all features is termed as "Basic LOSOCV RFR". However, training a RFR model requires least computational resources, thus eliminating features doesn't necessary. The entire manuscript compares Optimal vs Basic LOSOCV RFR models, which makes the manuscript more like a lab report, rather than a scientific article. The authors should come out of the thought of evaluating multiple RFR, but rather bring new novelty.

12. The authors should consider feature engineering, rather than merely feature importance.

13. Same as previous studies, this study also reports poor performance in LLJ cases. In fact, the RFR model predicted profiles doesn't even fall under LLJ cases, since no jet nose and increased/decreased shear below/above the nose are identified (though the authors did not show the RFR predicted profiles, it is evident from the bias plots). This further hinders the novelty of this study.

14. Sections 5.1 and 5.2 should be reduced with limited metrics in the text. Rather keep the metrics in the figures. Instead of reporting the metrics in quantitative way, explain the reasons behind the poor/better performance, which gives a scientific reasoning to the reader.

15. Figures 3-12: Increase the font size and make them clearer.

16. Tables 2-3: Comparing with the conventional Logarithmic Law and Power Law does not bring any new scientific insights, rather repeats what the previous studies have found.

17. The authors stressed that the RFR model can perform skillfully around the coast of the contiguous US, including regions not included in the training data. However, it is

paramount of importance to note that the RFR model doesn't know which locations the data comes from (since no lat/lon information are provided), rather it only knows the correlations within the data.

18. The authors spent vastly on validating the RFR model in comparison with conventional methods but spent very little on the generation of NOSP and the insights from this data. To make this study novel, conduct offshore wind resource assessment using NOSP, and several wind turbine models. A Levelized Coast of Energy (LCOE) using NOSP could be a novelty.

19. Lines 537-657: this analysis is not necessarily in the main manuscript, since the ERA5 and NOW23 are expected to be correlated, due to their parent/child relation. Move it to the Supplement.

20. The last paragraph of results seems be out of context. Could you elaborate why this was explained here?

Minor comments:

1. Limit abstract to one paragraph, only providing necessary overview, without describing in detailed.

2. Lines 134-135: the NOW23 is generated by using the WRF model, provided ERA5 reanalysis. Rewrite the sentence "this product implements the WRF ..."

3. At this point, the RFR model has been widely used in wind speed extrapolation. Thus, move the RFR model description to supplement, and only explain the training with detailed flowchart of inputs and targets.

4. Tables: Put the captions on top of the table.

---

## Author Comment (AC1)

**We thank the reviewer for their time and their responses to our manuscript. Our replies are inserted below into the reviewer's comments.**

This manuscript provides the methodology and analysis behind a newly released publicly available offshore wind dataset called NOST (NOAAOffshoreWindProfiles-USA). The data set is six-hourly at 0.25-degree resolution and spans from 1987 to 2022. The authors implement a machine learning method (random forest regression) to extrapolate wind profiles from satellite-derived surface winds (NOAA's National Centers for Environmental Information's Blended Seawinds version 2.0 (NBSv2.0) product). They train their extrapolation model using lidar data and validate it at the same lidar sites as well as at two other locations whose data was not used for training. They compare their wind profiles with reference wind profiles (log-law, power law) against observations and find that their model outperforms reference wind profiles (as expected). They quantify their errors and also analyze subsets of the data to focus on low-level jets and high-shear conditions. Lastly, they use a triple collocation method to compare the NOSP product with ERA5 and the NOW23 wind dataset and find the lowest estimated errors for NOSP regardless of coastal region. The paper is very well-written, and I believe their dataset would be highly valued by wind farm developers. Overall, this is a great paper that can be improved with a few minor revisions as suggested below.

**Specific Comments**

1. I would suggest removing or streamlining the first three sentences in the abstract. Considering that this is a wind energy-focused journal, the statements are already widely understood by the journal's target audience.
   **The first three sentences were removed.**

2. I do not think that both the log-law and power law need to be used as references. I would recommend just choosing one as they are going to be similar and with similar limitations.
   **While we agree that only using one model for comparison may suffice, we believe that comparing against both is more thorough and readers may wonder how our model compares to the law we drop. As such we would prefer to leave both in if the reviewer finds that acceptable.**

   Additionally, I could not find the surface roughness value used to generate the log-law profiles. Does the value vary spatially or temporally due to ocean conditions? The authors state that a lack of knowledge regarding u* and z_0 restrict them to only the neutral log-law, but you still need z_0 for the neutral log-law so I am confused why u* could not also be determined. I would also think that u* is also an output of ERA5 similar to the SST.
   **The surface roughness length used for the neutral log law is a constant z0 = 0.0001 (as in Optis et al., 2021). This has now been added into the paper (Line 180). As we used a constant z0, it would not be possible to calculate $u_*$ from z0 with any meaningful information as it would also be a constant.**
   **$u_*$ is also available from ERA5 but is unavailable at the exact location of lidar stations used to train the model. As ERA5 has lower spatial (0.25° vs. in situ) and temporal resolutions**

**(1 hour vs. 10 minute) than the lidar station wind profile data, using ERA5 u$_*$ could add more error into the log-law model when used for comparison at these stations. We use SSTs measured in situ at the lidar stations in the model training, not SSTs from ERA5. ERA5 SST and air temperature are only used in our analysis as RFR model inputs when extrapolating the Blended Seawinds dataset to full profiles as the two products have the same spatial resolution.**

3. In general, there are a lot of figures with vertical profiles. I would recommend a few things to improve readability As in 2, I would suggest removing either the log-law or the power law. RMSE and MAE are also similar metrics. I would recommend just showing one of the two.
**We agree that RMSE and MAE are similar metrics and have decided to remove RMSE from the paper. As for removing log and power law, please see our reply to comment #2 above.**

4. In general, the figures need to take advantage of the width of the page to improve readability. They are too small as they currently exist. They also need to be centered. The captions for figures 8 and 9 need to be on the same page as the image.
**Figures have been resized to fit the page better and centered. The captions for all images are now on the corresponding pages to the images.**

5. Did the authors consider changing the shear exponent for the power law for the higher shear cases? I assume that the authors want a model that is independent of parameters that should be tuned; however, using a power law with alpha=0.1 is obviously going to underestimate wind speeds higher up during high-shear conditions.
**The reviewer is correct that we wanted to compare against a model that is independent of tuning parameters. While alpha = 0.1 is obviously underestimating wind speeds during high shear conditions, a power law would have to calculate alpha in a way such that it knows when high shear conditions are occurring. This is not the case for most formulations we have seen, and the calculation we have seen for alpha that does take into account atmospheric stability and could potentially estimate wind during high shear cases also relies on stability correction functions that contain parameters unavailable from the lidar datasets.**

    **We also wanted to compare against models that could be used for the purpose of extrapolating winds at all grid points in our dataset, so calculating a different constant alpha for the purpose of high shear conditions specifically would not work as when applying the model across the whole region in our dataset as there would be no way to know beforehand which value to use without already knowing if the case is high shear or not.**

**Minor Comments**

- It would improve readability of Tables 2 and 3 if the data subset (overall, normal, etc.) was a separate column

**We agree with the reviewer and have added a new column for data subset "Profile Type" as a separate column.**

- Table 2: the entry on the second row for the MAE column is missing a '%'.

**Thank you for finding this. The missing '%' is now added.**

---

## Author Comment (AC2)

Reviewer 3 Response

This study presents a valuable attempt to address the scarcity of offshore wind speed profile data by developing a machine learning-based extrapolation model utilizing satellite-derived wind data, which could benefit wind resource assessment along U.S. coasts. The authors should be commended for their extensive data collection and model validation efforts across multiple coasts, showcasing the model's performance relative to traditional methods like the Logarithmic and Power Laws. However, while the work offers interesting results, there are notable areas where the study could be improved. Primarily, the claimed novelty of the research is somewhat diluted, as the Random Forest Regression (RFR) model has already been widely applied in similar extrapolation studies, and the spatial and temporal resolution of the resulting dataset is still coarser than other products like NOW23. Additionally, the reliance on standard stability assumptions and shear exponent limitations in the comparison methods may not fully reflect real-world conditions, and the leave-oneout cross-validation (LOSOCV) approach, though thorough, detracts from the goal of developing a single, generalizable model. Furthermore, the study would be strengthened by more focus on the practical application of the new dataset (NOSP) and its insights for wind energy stakeholders, potentially incorporating o shore wind resource assessments and Levelized Cost of Energy (LCOE) analyses. These enhancements, along with restructuring of certain sections and clarifications in the visual elements, could provide a clearer scientific contribution and highlight the value of the NOSP dataset for the wind energy community. After careful evaluation, I find that the manuscript, in its current form, lacks sufficient novelty to warrant publication. To enhance the paper's contribution, I recommend the authors consider incorporating novel elements that extend beyond current literature, such as using the derived dataset (NOSP) for offshore wind resource assessments or cost analyses (e.g., LCOE calculations) that can directly benefit the wind energy industry. If such advancements are added, the work would more clearly demonstrate its unique value. Specific comments:

**We appreciate the reviewer's comments and the effort they put into reviewing our manuscript. We understand that the reviewer has concerns and hope that our comments will address these and help clarify the novelty of our work.**

1. With the current pace of growing wind turbine sizes, it is of paramount importance to accurately measure the wind speed profiles, rather than just hub height wind speed. Introduce the importance of wind speed profiles, in terms of their interaction with wind turbines and their impact on loading and power estimation.

**Changes have been made to include mentioning the importance of measuring wind speeds within the rotor layer of turbines in addition to just the hub heights throughout the introduction (lines 43–46, 58). The importance of wind speed profiles and impacts on wind energy production, especially of LLJs and high shear events, are also included in the introduction and beginning of section 5.2 (lines 84–87, 342-343).**

2. In addition to tall towers and lidars, sodars also serve as wind profile measuring instruments. Introduce them. Also, lidars have several limitations, such as failed to measure wind speed during raining, snowing, or other precipitation events; no measurement without aerosols. Instead of "lidars are very accurate alternative devices", at lines 54, introduce the limitations as well.

**We have rephrased calling lidars a "very accurate" alternative and instead consider them an alternative with a few limitations listed (Lines 51–54). A couple of sentences on sodars are also included in the the introduction (Lines 47–51)**

3. Line 59: 20 years of mesoscale model simulated hindcasts of wind profiles (NOW23) can offer a concise resource assessment, which are sufficient in temporal scale, covering annual to decadal analysis, and at a high resolution of 2 km. It doesn't appear to be a gap in the long-term wind speed profile knowledge along the US coast. Rephrase this sentence.

**The sentence has been rephrased to focus more on the gap in real-time wind speed knowledge only (lines 59–60).**

4. Lines 60-64: The authors claim the novelty lies in the idea of bridging critical database gap. However, the NBSv2.0 gridded dataset available at a resolution of 0.25 deg, and at a temporal resolution of 6 hours, which is poorer in temporal and spatial resolution, than NOW23, except for the coverage. Here it dilutes the novelty.

**We recognize the dataset is at poorer spatial and temporal resolutions compared to NOW-23. The NOW-23 dataset is a highly valuable dataset for wind energy developers, especially for its high resolution. However, our dataset differs in a few important ways that address the unmet needs of some stakeholders. Our wind profile dataset provides observations from 1987 to present (currently 38 years) compared to NOW23's ~20 years (from 2000 to 2019–2022; end date depends on region). Our dataset will have near real time production with ~1 day delay and monthly processed science quality data, allowing wind developers access to the most recent wind profile observations, compared to NOW23 which is not regularly updated and does not contain data for the most recent years. We realize we had not included all these details in the initial manuscript submission and had said 1987–2022 instead of 1987–present for the time span of our product. This has been corrected in the manuscript where appropriate (lines 16, 23, 33, 64, 101, 485, 627).**

**Further, our dataset incorporates satellite observations and uses the random forest model to estimate wind profiles, compared to NOW23 that is solely based on ERA5 reanalysis and a WRF model. Comparatively, NOAA's Blended Seawinds blends observations from multiple satellites and much more accurately estimates high wind speeds, including extreme weather events, compared to reanalysis data including ERA5 (https://www.frontiersin.org/journals/marine-science/articles/10.3389/fmars.2022.935549/full). The WRF model implemented in NOW23 is based on the logarithmic law for profile extrapolation, which has been shown in many studies (including several cited in our introduction) to be outperformed by the random forest. These results are validated for our product by the triple collocation analysis. Our dataset shows lower errors compared to those of ERA5 and NOW23 around the US coastal regions. While we do understand the value of both ERA5 and NOW-23 to many users, we believe that the addition of our dataset would further benefit stakeholders in the offshore wind energy industry. These two datasets complement each other, together providing users more comprehensive information.**

5. There are several studies extrapolated wind speed from surface to 200 m level, utilizing random forest regression (RFR). A very recent study under review is Rouholahnejad et al. (2024), which did exact similar work. The authors could have referred to this article. In addition, the RFR methodology has been adopted in several other studies, showcasing the better predictability in low-level jet cases and high

shear events, compared to the conventional low-logarithmic law power law. In comparison to the existing studies, there seems little added value in this study to the scientific community. Rouholahnejad, F. and Gottschall, J.: Characterization of Local Wind Profiles: A Random Forest Approach for Enhanced Wind Profile Extrapolation, Wind Energ. Sci. Discuss. [preprint], https://doi.org/10.5194/wes-2023-178, in review, 2024.

**The authors were unaware of this study at the time of our submission. While we understand that this study's title suggests it may overlap with our analyses, upon further review it seems that it contains significantly different analysis than the work we have presented in this manuscript. The Rouholahnejad and Gottschall study, similar to many previous studies we have already referenced in the Introduction (Lines 72–97), focused on validating the model in a small region, at stations within 200 km of each other. They also noted that while their model captures regional effects well, the bias increases at locations 200 km away from the training data. The limited ability of their model for profile extrapolation far from the training data locations is a sharp contrast to the work presented here where we develop an RFR model, through extensive hyperparameter tuning and feature selection using cross validation, that generalizes well and does not overfit to one small region. This is what enables us to produce a wind speed product for all the coastlines of the contiguous US and Hawai'i using this model. By testing our resulting wind speed estimates against the two ASOW and several NOW-23 stations, we find that our RFR model is able to robustly produce such estimates in a variety of regions substantially farther than 200 km from our training data locations. We are not aware of any previous studies, including the Rouholahnejad and Gottschall study, that developed an RFR model for extrapolating wind speeds that could generalize well to multiple coasts and in distant regions (relative to the training data locations). Furthermore, we are not aware of a study that introduces a long-term satellite observation based and gridded dataset of wind speed profiles created using the random forest model, as we have done in this work. We hope this helps clarify the major differences between the Rouholahnejad and Gottschall study and this work.**

6. In fact, this current study resembles extension of Optis et al., 2021. However, there is limited scientific addition in novelty.

**See reply comment on #5. Additionally, The study from Optis et al., 2021 focuses on two of the Hudson buoys which are 83 km apart for their analysis. As such, we cannot assume that their RFR model will work all around the US coasts without performing that analysis ourselves as their model could have learned localized climate patterns to help in extrapolating profiles that would not necessarily be the same at all offshore locations. We innovate from the Optis et al., 2021 model by training our RFR using more lidar stations and doing a more detailed feature selection process. Furthermore, we perform a more rigorous assessment of our model's ability to generalize to unseen locations through comparisons to the two ASOW stations and additional NOW-23 stations, while also comparing its performance to the ERA5 and NOW-23 reanalyses by performing a correlated triple collocation analysis. Further, Optis et al., 2021 does not introduce a dataset based on the random forest model as we have done here.**

7. Lines 94-96: the authors state that the main novelty of this present work lie in extrapolating wind speed profiles on a larger spatial scale covering multiple coasts. However, they focused vastly on the RFR model and its validity compared to the conventional methods, which in fact several researchers have

already reported. Similar to the previous studies, the current study also failed to accurately estimate the LLJ cases. If the authors have captured the LLJ events, it could be a stand-alone novelty.

**Our responses to earlier comments (#5 and #6 above) addressed this comment as well. Our focus on the model and its validity against conventional methods is to ensure that it is clear that we have examined that this specific RFR model is capable of extrapolating wind speeds with higher accuracy than other models across large spatial scales. While other papers have validated the RFR model in regional analyses, no paper has validated across as large of a region as done here (contiguous US and Hawai'i coastlines). Thus, it is imperative for us to demonstrate the model is capable of extrapolating wind speeds with higher accuracy compared to conventional methods over such a broad region before using it to generate a product covering the coastal US regions.**

**We recognize that the inability of our model to currently capture LLJ events is a shortcoming of the model. We are in the process of furthering this work to capture the LLJs in a future version of the product and a future paper. However, given that >94% of profiles at each station are non-LLJ profiles, we feel our current model is able to perform well at replicating the majority of observed wind profiles.**

8. Lines 143-150: this paragraph seems out of order and introduces confusion, since the ML methodology has not been explained. For a clarity, remove this from here and explain it when introducing the RFR model training.

**This paragraph has been removed and it is explained instead in the sections for their given analyses (Lines 151, 458–460, 524).**

9. One major drawback of this study is using the conventional Logarithmic Law and Power Law for comparison, which have been proven to be inadequate in extrapolating wind profiles, due to their dependance on stability conditions and shear exponent. Also, the assumption of neutral conditions for Logarithmic Law and a constant sear exponent for Power Law already introduces substantial errors in wind profile extrapolation. To make the study novel, it is better to consider all the stability conditions and variable shear exponent in estimating wind speed profiles.

**Due to not having measured friction velocity or surface roughness sampled at the same resolution as the SST and w10 at the lidar stations, we were not able to consider stability correction functions. One of the strengths of our model is its ability to predict high wind shear, which generally occurs during stable conditions, as seen in Figure 8 where most high shear cases have a positive air-sea temperature difference. On the other hand, the logarithmic law is known to break down under stable conditions and is incapable of predicting LLJs. This was shown in Optis et al., 2021 which showed the logarithmic law greatly underestimating wind speeds in stable conditions at Hudson North and Hudson South buoys. As such, we can assume that the logarithmic profile (at these two buoys at least) breaks down in cases of high wind shear whereas our model is capable of predicting these high shear cases in stable conditions. Since the logarithmic law is known to break down in stable conditions where our model shows to perform well (excluding the inversion in LLJs), we can assume our model would similarly outperform the logarithmic law considering stability as well. Again, in addition to the methodology, here we also create a gridded, near-real-time wind profile product based on the random forest model and multiple satellite based observation, showing our model can perform skillfully across large regions.**

10. The authors gave a tremendous importance to the leave-out-station-out cross validation. It is common practice to split the data into training and validation, to optimize the model parameters. Instead, the authors constructed five RFR models by leaving one station at a time, and testing the model accuracies on these stations. This actually comes into "testing" the model accuracy, but not the "validation". The authors should understand that the wind energy community surpassed validating the accuracy of RFR model, rather a unique model accurate enough to extrapolate the entire wind speed profile is needed.

**We thank the reviewer for your comment, however, we feel there is some confusion about our use of LOSOCV, which was in fact used for the training and validation process only. Specifically, the use of LOSOCV in this manuscript is to optimize the hyperparameters and select features that will be used for the final model. The features are selected by assessing the increase/decrease in error on the held-out stations. This point is now further elaborated in the revised manuscript (lines 260, 268–269, 272). The hyperparameters were tuned using the out-of-bag training error, however the optimal values were still validated on the held-out locations to ensure they generalize well to data at distant locations. This approach provides a more detailed way of selecting optimal hyperparameters than only using one validation set since only holding out a single location as the only validation set can lead to overfitting a model that might only be optimal at that location. For example, one input may show an increase in model accuracy at Humboldt when added, but a decrease in accuracy at Morro Bay when added. Our analysis showed the optimal values for the hyperparameters did not change depending on the held-out location, which we could not have proved by only using one training/validation split alone. As we used the hold out stations to select features and determine improvements from hyperparameters, as well as the fact that all five stations are used as training data in construction of the final model for our product, none of the steps involving the LOSOCV models can fall into the category of "testing" since they all contributed to model development. This is why the two Atlantic Shores wind stations are introduced as testing stations. Neither station was used at all in the training and validation of hyperparameters and feature selection so they provide data independent of our model training to test the model's ability to generalize to unseen data. The NOW-23 stations serve a similar purpose.**

**While we have seen other papers from the offshore wind community validating the use of the random forest, there are no studies we are aware of that validate a model over as large of a region as used here. Other papers have focused only on validating the model in a specific small region, including Rouholahnejad and Gottschall (2024) which focused on the North Sea only. While the fact that regional models have been well validated is promising for a model to be used over such a large region, it does not necessarily mean that fact can be taken for granted. In fact, our LOSOCV shows that the LOSOCV RFR with all features input into the model had slightly lower errors at the three Hudson stations while performing worse than the optimal RFR at other stations. This points to additional (non-w10 or ΔT) features potentially having more feature importance in models used in small regions, consistent with Optis et al. (2021) (at the Hudson stations) and Rouholahnejad and Gottschall (2024) (in the North Sea). Here, we aimed to train and validate a model that is optimal for extrapolating winds over a large scale and generalizes well to many areas, which had not been done before.**

Instead of bombarding the manuscript with LOSOCV model, the authors should bring new insights from the one single optimized RFR model.

**Thank you for this comment as it highlighted the need for us to clarify in the manuscript when the five LOSOCV RFR models were being used as opposed to the final, optimal RFR trained on all 5 training/validation stations. In addition to the existing note on line 296, we have made this more clear at the start of sections 6.1 (line 421) and 7 (line 484). We provide insights into how the one single optimized RFR performs showing that it is capable of extrapolating winds at locations completely independent of the training/validation stations, including independent model testing at the Atlantic Shores stations (lines 420–455) and 6 different NOW-23 stations (lines 458–481) in regions far from the training data, such as two stations in the Gulf of Mexico where no training data is provided. Further, this single optimized model is applied to the NOAA Blended Seawinds dataset to produce our new wind profile product and error analysis is done on the resulting product and compared to ERA5 and NOW-23 (lines 484–563). The error analysis shows our dataset created with the final optimized random forest model outperforms both ERA5 and NOW-23 with the lowest errors among the three datasets. No other paper we are aware of has applied a random forest model to a gridded satellite-based dataset for widespread wind speed profile estimation nor compared the resulting profiles with other current wind profile reanalysis datasets.**

11. The authors conducted feature importance and came up with only two variables, namely 10 m wind speed and temperature difference. The model trained with these parameters is termed as "Optimal LOSOCV RFR", while the model trained with all features is termed as "Basic LOSOCV RFR". However, training a RFR model requires least computational resources, thus eliminating features doesn't necessary.

**While eliminating features may not be necessary to significantly reduce computation time, including features that do not improve the model is known to degrade the RFR's performance because they are, in a way, adding noise to the model. Hyperparameter tuning can often resolve this issue to a degree but we decided to take a more direct approach to limit the impact of these less predictive variables in this study. We also found that including some features actually decreased accuracy at some held out locations, so it is important that those be removed.**

The entire manuscript compares Optimal vs Basic LOSOCV RFR models, which makes the manuscript more like a lab report, rather than a scientific article. The authors should come out of the thought of evaluating multiple RFR, but rather bring new novelty.

**The comparisons between the optimal and basic LOSOCV RFR are limited to the model training/validation sections while the rest of the manuscript's analyses use the optimal RFR. We have strived to make it more clear in the manuscript when which of these models are being used (see comment #10 reply above). The manuscript also tests the accuracy of the optimal model on independent testing locations at Atlantic Shores Offshore Wind 4, Atlantic Shores Offshore Wind 6, and six NOW23 locations as independent testing from our training locations is needed. This independent testing demonstrates that the optimal model can accurately estimate wind speed profiles in distant regions from the training data where the model has no prior knowledge of wind resources, something that we could not find previously demonstrated in the literature. Furthermore, the manuscript not only evaluates the model's performance rigorously but also uses it to generate a gridded product around the whole US coastal regions, which has not been previously done. The manuscript then compares the random errors of our new dataset based on satellite data and machine**

**learning to ERA5 and NOW23 and shows lower random errors in all US coastal regions compared to ERA5 and NOW23.**

12. The authors should consider feature engineering, rather than merely feature importance.

**The authors believe feature engineering and feature importance to serve two different functions. We define feature engineering as transforming data into features useful for machine learning models. We did this for several of the candidate features for the RFR, including: (1) breaking down all the cyclical features in the manuscript into their sine and cosine components so that the model could capture their cyclical nature; and (2) calculating $\Delta T$ from the measured air temperature and SST. Given the authors' understanding of feature engineering, we believe we have already undertaken it as part of this manuscript.**

**On the other hand, we define feature importance as a measure of how much a feature improves predictions in an ML model. As such, feature importance was used to assess which features to keep, including features that were engineered from the raw data.**

13. Same as previous studies, this study also reports poor performance in LLJ cases. In fact, the RFR model predicted profiles doesn't even fall under LLJ cases, since no jet nose and increased/decreased shear below/above the nose are identified (though the authors did not show the RFR predicted profiles, it is evident from the bias plots). This further hinders the novelty of this study.

**We recognize that our model is not capturing LLJs and plan to address this in future research. We have mentioned that in the manuscript (lines 597–598) and discussed this issue in more detail in our reply to comment 7.**

14. Sections 5.1 and 5.2 should be reduced with limited metrics in the text. Rather keep the metrics in the figures. Instead of reporting the metrics in quantitative way, explain the reasons behind the poor/better performance, which gives a scientific reasoning to the reader.

**Sections 5.1 and 5.2 were rewritten to be more qualitative in explanations without as many values of the metrics placed in the text.**

15. Figures 3-12: Increase the font size and make them clearer.

**We made the figures bigger and increased the font sizes.**

16. Tables 2-3: Comparing with the conventional Logarithmic Law and Power Law does not bring any new scientific insights, rather repeats what the previous studies have found.

**The comparisons with the logarithmic law and power law is done to make sure the RFR still outperforms them over a broader region before implementing the RFR across such a wide swath of coastlines. While not a completely new idea in itself to validate against these methods, it does confirm the model's ability to accurately estimate wind speeds at far distances, which has not been confirmed as other studies on the model are all performed in small regions.**

17. The authors stressed that the RFR model can perform skillfully around the coast of the contiguous US, including regions not included in the training data. However, it is paramount of importance to note that the RFR model doesn't know which locations the data comes from (since no lat/lon information are provided), rather it only knows the correlations within the data.

**We agree that this is an important point to note. This shows that the model generalizes well to many offshore regions and is not reliant on lat/lon to perform well, instead drawing solely upon the relationship between w10 and ΔT to make the wind speed profile estimates. We have added this fact into the manuscript when talking about our inputs used to avoid any potential confusion for readers (lines 230–232).**

18. The authors spent vastly on validating the RFR model in comparison with conventional methods but spent very little on the generation of NOSP and the insights from this data. To make this study novel, conduct offshore wind resource assessment using NOSP, and several wind turbine models. A Levelized Coast of Energy (LCOE) using NOSP could be a novelty.

**We intended this manuscript to focus on the methodology and creation of our dataset as well as assessment of the uncertainty in our product. We would like to contribute further research using this data such as including it in wind turbine models and LCOE analyses. However, given the focus and length of the current manuscript we feel such additional analysis is beyond the scope of the current work. We thank the reviewer for the suggestions.**

19. Lines 537-657: this analysis is not necessarily in the main manuscript, since the ERA5 and NOW23 are expected to be correlated, due to their parent/child relation. Move it to the Supplement.

**Details of the correlation between ERA5 and NOW-23 and the bivariate plot were moved to the appendix as we believe it is more suitable there per WES standards. We still believe it is important to introduce triple collocation as background for correlated triple collocation and have kept some of the salient background discussion in the main text.**

20. The last paragraph of results seems be out of context. Could you elaborate why this was explained here?

**This paragraph was included to reiterate the fact that the error statistics that we obtained at 100 m altitude are slightly higher from what Bodini et al 2024 reported at 140 m.**

Minor comments:

1. Limit abstract to one paragraph, only providing necessary overview, without describing in detailed.

**The abstract is now shortened to one paragraph.**

2. Lines 134-135: the NOW23 is generated by using the WRF model, provided ERA5 reanalysis. Rewrite the sentence "this product implements the WRF …"

**This sentence has been rewritten.**

3. At this point, the RFR model has been widely used in wind speed extrapolation. Thus, move the RFR model description to supplement, and only explain the training with detailed flowchart of inputs and targets.

**We have found in our discussions with the community that briefly reviewing the general structure of the RFR before diving into its specific application here is useful despite its increase in prevalence. Given that some readers may find this brief review helpful and that the description is only a paragraph**

**in length, we think it would be more beneficial for it to remain in the text where it is more easily accessible to the reader. However, we also added a flowchart as suggested by the reviewer (see Fig 1).**

4. Tables: Put the captions on top of the table.

**Table captions are now moved to the top of the tables.**

---

## Author Comment (AC3)

Reviewer 2 Response

**We thank the reviewer for their time and their responses to our manuscript. Our replies are inserted below into the reviewer's comments**

The paper demonstrates the use of random forest regression (RSR) to estimate wind speed profiles based on satellite 10m winds and ERA5 temperature data (SST and 2m). The work is novel and the comparison is quite extensive comparing with several independent sources. My main concern is that the authors compare their results to a neutral log profile and a power law profile with a fixed exponent. The neutral log law is used as the authors state that there is insufficient information to make the stability correction using a diabatic profile. This seems somewhat surprising. They are using ERA5 as an input to their RSR model which already uses delta T. This could equally be used along with wind speed to infer a bulk Richardson number which could be related to Obukhov length (L) and a formulation of the psi(z/L) function such as that by Businger and Dyer used to make the stability correction to the neutral log law. Indeed other parameters from ERA5 (e.g. sensible heat flux) could be used to get a more accurate estimate of surface stability. This would seem to be a much fairer comparison than using the neutral log law.

**Thank you for the comments. In response to the concern about our use of the neutral log law, we would have preferred to use stability corrections. However, we are only comparing our RFR with the log law at the locations of the lidar buoys. At these stations, we do not make any use of ERA5 for SST since the buoys already provide in situ measurements of SST and air temperature (and thereby ΔT). Furthermore, ERA5's 0.25° grid is not always co-located with the buoy locations and ERA5 has lower temporal resolution (1 hour) than the lidar buoys (10 minute resolution). Our use of ERA5 for ΔT only comes in as an input into the RFR model when it is applied to extrapolate the Blended Seawinds 10 m winds speed to full profiles. The Blended Seawinds product is on the same 0.25° grid as ERA5.**

Furthermore, it would be interesting to see a more detailed assessment of the accuracy of the RSR/log law/power law profile as a function of direction to see if the coastal transition plays a role in model accuracy.

**We agree that this would be an interesting analysis. In this initial analysis we just focused on speed, however we could look into direction as well in future work.**

Finally, the results of the performance metrics for the floating lidar sites would be more readable as a table as was done for the ASOW sites.

**We agree that our initial discussion of the performance metrics for the float lidar sites was a bit dense. As part of a response to another reviewer, we revised sections 5.1 and 5.2 to not include as many values of the metrics. The values for these metrics are still shown on the figures in those sections. We believe this should resolve the concerns noted here.**

Minor comments:

- The value of z0 for the log law extrapolation does not seem to have been mentioned (unless I miss it somewhere). Or was a Charnock relationship used?

**The value of z0 used is 0.0001 (as in Optis et al., 2021). We also considered 0.0002 which has been assumed in offshore environments, but found 0.0001 to be more appropriate. This value has been added to the paper (Line 180) as the reviewer is correct that it was missing.**

- Units are missing in Table 4.

  **Thank you for finding this. Units have been added to the caption of table 4.**

- On page 8, the word 'importances' is used several times. This sounds odd and I suggest that 'importance' is used as a collective noun.

  **We agree with the reviewer that "importance" sounds better than "importances" and have made the changes necessary.**

- Line 522: change 'shown decrease' to 'show a decrease'

  **The phrase has been corrected.**